# Conditional Guided Flow Matching: Modeling Prediction Residuals for Enhanced Time Series Forecasting

## Abstract

Time series forecasting predominantly focuses on modeling the mapping between historical and future sequences, and existing improvements are often constrained to optimizing model architectures to better capture this relationship. This essentially reduces prediction residuals to mere optimization targets while overlooking their informative structures such as systematic biases or nontrivial distributions that could otherwise be exploited to directly reduce forecasting errors. Unfortunately, discriminative models struggle to capture the complete residual structure and its dynamic temporal dependencies when applied to residual learning. To fill this gap, we introduce Conditional Guided Flow Matching (CGFM), a novel framework built upon flow matching. CGFM innovatively leverages auxiliary predictions as the source distribution and constructs two-sided conditional paths to prevent path crossing, which enables the explicit learning of the full structure of prediction residuals and thereby theoretically guarantees superior performance over discriminative models. Extensive experiments show that CGFM enhances diverse forecasting models including state-of-the-art ones and demonstrates its effectiveness and generality. Code link: `https://anonymous.4open.science/r/CGFM-31DB`.

## 1 Introduction

Time series forecasting, a fundamental task in time series analysis, has broad applications and significant impact in domains such as finance, healthcare, and energy (Lim & Zohren, 2021). The core objective is to learn the mapping from historical sequences to future sequences. Existing models (Chen et al., 2025; Huang et al., 2025; Wang et al., 2025b) improve performance by refining architectures to minimize the residuals between predictions generated from historical sequences and the ground-truth future sequences, thereby better capturing the underlying temporal relationship.

However, this formulation treats prediction residuals merely as optimization targets, overlooking their informative structures—such as systematic biases or nontrivial distributional patterns. This raises a natural question: **Can we explicitly learn from residuals to directly correct the errors of forecasting models?** Intuitively, if we can effectively model residuals, adding the learned residual corrections back to the original predictions would refine the forecasts and enhance accuracy. A straightforward idea is to use a discriminative model for residual learning. Yet both our theoretical analysis and experimental results (Appendix A.2) demonstrate that such models are mathematically constrained to fitting only first-order moments (conditional means) and struggle with integrating historical context, leading to poor performance in residual learning.

To fill this gap, we draw on the strengths of generative models. These models (Kollovieh et al., 2025; Li et al., 2025; Shen & Kwok, 2023b; Tashiro et al., 2021) have emerged as powerful tools for time series forecasting, equipped with prediction distribution modeling capabilities and conditional generation mechanisms. We build on flow matching (Esser et al., 2024; Kerrigan et al., 2023; Lipman et al., 2023), a more flexible generative framework featuring adaptable initialization and sampling paths. We develop Conditional Guided Flow Matching (CGFM), which enhances time series forecasting by leveraging auxiliary predictions as the initial distribution to explicitly model the complete residual structure, refines forecasting results, and theoretically guarantees superior performance over discriminative models (Appendix A.2).

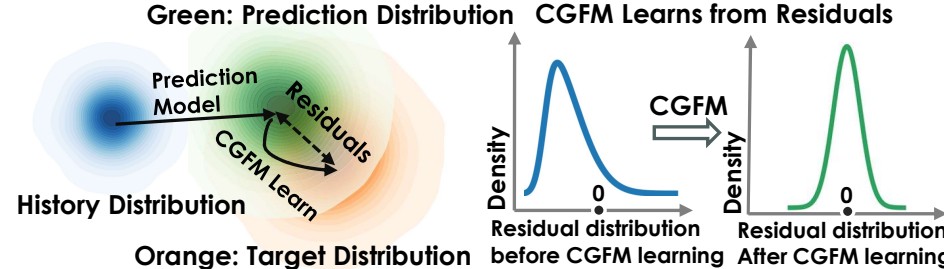

Figure 1: Left: Since no model can predict with perfect accuracy, their prediction distribution inevitably differs from the target, leading to residuals. CGFM learns the probabilistic structure of these residuals. Right: CGFM reduces the residuals to be centered and concentrated around zero (mean tending to zero with significantly reduced variance).

First, we utilize the distribution of a prediction model's predictions as the source distribution, rather than constraining it to simple noise. As illustrated in Figure 1, CGFM learns a transformation path from the prediction distribution to the target distribution. We have proven (Proposition 4.2) that this formulation integrates the explicit modeling of the full residual structure into the flow matching framework, enabling end-to-end acquisition of enhanced forecasts directly. At the same time, using the auxiliary prediction distribution as the source in flow matching not only preserves richer temporal dependencies than a Gaussian prior but also provides the closest accessible approximation to the target distribution, thereby reducing the difficulty of residual learning.

Second, the essence of forecasting lies in capturing the mapping from historical sequences to future sequences. To this end, we incorporate historical information into both probability path construction and velocity field learning. A key innovation is a two-sided conditional probability path, in which both the source and target distributions are conditioned on the same historical data. This design, combined with our choice of source distribution, model structure, and the flexibility of affine paths, theoretically guarantees the avoidance of path crossing—a critical challenge in flow-based modeling (Tong et al., 2024; Liu et al., 2022b). By preventing path crossing, our framework reduces prediction ambiguity and information loss during sampling, which enhances forecasting accuracy. Historical data also guides the velocity field to capture temporal dependencies, improving the alignment between source and target distributions. In addition, reparameterizing the prediction target to directly optimize toward the ground truth yields a further accuracy gain.

Our main contributions are as follows:

- We propose a novel formulation for time series forecasting that goes beyond simply minimizing prediction errors to learning from residuals. Backed by theoretical analysis and proofs, our framework outperforms discriminative models by explicitly capturing the full structure of prediction residuals, thereby refining the auxiliary model's forecasts.

- We theoretically adapt flow matching to the time series domain and propose CGFM, which integrates historical data as two-sided conditions in both path construction and velocity field learning. This theoretically grounded design avoids path crossing, enhances temporal dependency capture, and achieves further gains through target reparameterization

- We conduct extensive experiments, demonstrating that CGFM consistently improves forecasting performance validating its effectiveness and generality.

## 2 PRELIMINARIES

### 2.1 FLOW MATCHING

*Note: The notations in this section are for theoretical illustration and are not the same as those used for time series forecasting.* Given a sample $X_0$ drawn from a source distribution $p$ such that $X_0 \sim p$, in $d$-dimensional Euclidean space where $X_0 = (x_0^1, \ldots, x_0^d) \in \mathbb{R}^d$, and a target sample $X_1 = (x_1^1, \ldots, x_1^d) \sim q$. Flow Matching (FM) constructs a probability path $(p_t)_{0 \leq t \leq 1}$ from the known distribution $p_0 = p$ to the target distribution $p_1 = q$, where $p_t$ is a distribution over $\mathbb{R}^d$.

Specifically, Flow Matching employs a straightforward regression objective to train the velocity field neural network, which describes the instantaneous velocities of samples. The relationship between the velocity field and the flow is defined as:

$$\frac{d}{dt}\psi_t(x) = u_t(\psi_t(x)), \tag{1}$$

where $\psi_t(x)$ represents the flow at time $t$, and $\psi_0(x) = x$. The velocity field $u_t$ generates the probability path $p_t$ if its flow $\psi_t$ satisfies $X_t := \psi_t(X_0) \sim p_t$ for $X_0 \sim p_0$. The goal of Flow Matching is to learn a vector field $u_\theta(t)$ such that its flow $\psi_t$ generates a probability path $p_t$ with $p_0 = p$ and $p_1 = q$. The Flow Matching loss is defined as: $\mathcal{L}_{\text{FM}}(\theta) = \mathbb{E}_{t,X_t}\left[\left\|u_t(X_t) - u_t^\theta(X_t)\right\|^2\right]$.

## 3 RELATED WORK

Flow Matching (FM) has gained attention in time series modeling for efficiently constructing continuous probability paths via velocity fields. Early work like CFM-TS (Tamir et al., 2024) explored FM for generation using Brownian bridges and Gaussian processes, offering better stability than neural ODEs. Later methods, however, share critical drawbacks: TSFlow (Kollovieh et al., 2024) just uses historical data during inference (weak historical utilization), FM-TS (Hu et al., 2024) lacks structured conditioning, and TFM (Zhang et al., 2024) relies on general priors and historical window constraints (underutilizing temporal dependencies). Collectively, these reflect two core limitations: weak historical data usage and reliance on generic priors. Treating the auxiliary prediction model and CGFM as an integrated forecasting model advances time series forecasting by resolving these issues. CGFM's two-sided conditional design ensures temporal consistency, efficiency and flexibility. It innovatively leverages the auxiliary model's predictive distribution, which is inherently closer to the target than Gaussian priors, to learn and correct residuals. Its core innovation is two-sided conditional paths with affine interpolation, explicitly guided by historical data to capture complex residual dynamics, avoid path crossing and boost predictive accuracy. And CGFM, through the design of its probability path, theoretically and naturally ensures non-crossing paths in time series tasks, which facilitates sampling and prediction—addressing a key challenge in flow matching (Zhang et al., 2025; Tong et al., 2024).

## 4 METHODOLOGY

### 4.1 PROBLEM FORMULATION

In time series prediction tasks, the goal is to leverage historical data to predict future data. Let $H \in \mathbb{R}^{C \times L} \sim p_H$ denote the historical data, with samples represented by $h$. Similarly, let $F \in \mathbb{R}^{C \times F} \sim q$ represent the target future data, with samples denoted by $f$. The probability path is defined as $p_t$, where $X_t \sim p_t$ represents the state of the data at time $t$. The objective is to generate more accurate predicted data that closely approximates $F$. Specifically, at $t = 1$, $X_1 \in \mathbb{R}^{C \times F} \sim q$, with samples denoted by $x_1$. At $t = 0$, the source data is given by $X_0 \in \mathbb{R}^{C \times F} \sim p$, with samples represented by $x_0$. Fig. 2 provides an overview of the CGFM framework

### 4.2 TWO-SIDED CONDITIONAL GUIDED PREDICTION

In time series forecasting tasks, for a given $h$, there exists a correspondence between the samples of the source distribution and the target distribution. Relying solely on one-sided conditioning, as in previous flow matching methods, is therefore inadequate. To address this, we propose two-sided conditionally guided probability paths, where a marginalization probability path is constructed and integrated to obtain $p_{t|H}(x|h)$: $p_{t|H}(x|h) = \int p_{t|0,1,H}(x|x_0,x_1,h)\pi_{0,1|H}(x_0,x_1|h)dx_0dx_1$. Here, $\pi_{0,1|H}(x_0,x_1 \mid h) = q(x_1 \mid h)p(x_0 \mid h)$, indicating that $x_0$ and $x_1$ are independent given $h$, a concept referred to as conditional independent coupling. Both are related to the historical data $h$.

The two-sided conditionally guided probability path is required to comply with the boundary constraints $p_{0|0,1}(x|x_0,x_1,h) = \delta_{x_0}(x)$ and $p_{1|0,1}(x|x_0,x_1,h) = \delta_{x_1}(x)$. Here, $\delta$ denotes the Dirac delta function. Subsequently, the velocity field can be obtained as: $u_t(x|h) = \int u_t(x|x_0,x_1,h)p_{0,1|t,H}(x_0,x_1|x,h)\,dx_0dx_1$. By Bayes' Rule, it follows that:

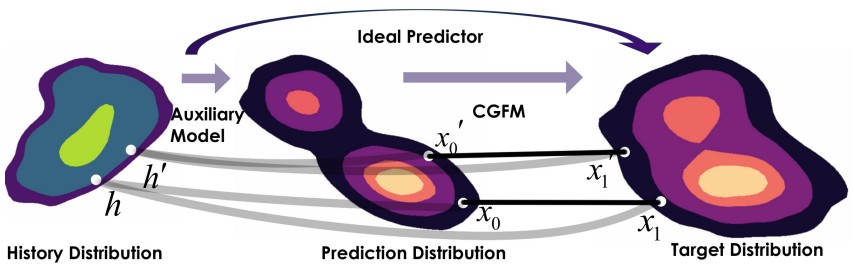

Figure 2: Visualization of CGFM Training. Prediction distribution deviates from target (no model is ideal), so CGFM focuses on learning prediction residuals. Black line: two-sided conditional path between $x_0$ (auxiliary prediction) and $x_1$ (ground-truth); gray line: historical data $h$ that generates $x_0$ and $x_1$. Given $h$ and $x_0$, CGFM outputs the enhanced prediction $x_1$.

$$p_{0,1|t,H}(x_0, x_1|x, h) = \frac{p_{t|0,1,H}(x|x_0, x_1, h)\pi_{0,1|H}(x_0, x_1|h)}{p_{t|H}(x|h)}. \tag{2}$$

Thus, the model learns $u_t(x|x_0, x_1, h)$ to obtain $u_t(x|h)$. According to Eq.(1), $u_t(x|x_0, x_1, h)$ determines $p_{0,1|t,H}(x_0, x_1|x, h)$, and vice versa.

## 4.3 CGFM AND RESIDUAL LEARNING

To fully leverage the valuable information encoded in the prediction residuals, we harness the flexibility of flow matching by setting $X_0$ as the output generated by a predictive auxiliary model, i.e., $X_0 = X_{aux} = \Phi(H)$. In this case, the distribution $p$ conditioned on $h$ can be expressed as $p(x_0 \mid h)$, representing the distribution of the auxiliary model's predictions. Target distribution $q$ used during training is derived from the inherent correspondence between future data $F$ and historical data $H$ in the dataset, which can be formulated as $q(x_1 \mid h)$. Specifically, we design a probability path that from the distribution of the auxiliary model's predictions to the target distribution. This allows flow matching to effectively learn from prediction residuals of the auxiliary model.

Since time series datasets are typically stored with a limited number of significant digits and consist of a finite number of samples, even if the original time series is continuous, $H$ can only be considered to have an approximately continuous distribution in $\mathbb{R}^{C \times L}$. In general, most models are differentiable, and thus $\Phi(H)$ can also be approximated as $C(\mathbb{R}^{C \times F})$. However, this is insufficient to ensure that $p(x_0 \mid h)$ satisfies the $C^1$ smoothness condition in $\mathbb{R}^{C \times L}$. This smoothness is a foundational premise for justifying the flow matching construction, as formalized in Lemma 4.4. We therefore use Gaussian noise to smooth $p(x_0 \mid h)$.

**Proposition 4.1** (Noise Smoothing). *Let $X \in \mathbb{R}^{C \times F}$ be a time series with distribution $P_{ori}$. Define the perturbed series $R$ as: $R = X + \sigma\epsilon$, where $\epsilon \sim \mathcal{N}(0, I)$ is additive Gaussian noise and $\sigma > 0$. The perturbed series $R$ follows the distribution $P_{per}$:*

$$P_{per}(r) = \int P_X(r - \sigma\epsilon)P_\epsilon(\epsilon) \, d\epsilon, \tag{3}$$

*which belongs to $C^1(\mathbb{R}^d)$. Furthermore, $P_{per}$ has a strictly positive density and possesses finite second moments. The proof is given in Appendix A.3.*

Intuitively, $P_{per}$ can be regarded as the convolution of the original distribution $P_X(x)$ with a Gaussian distribution $P_\epsilon(\epsilon)$. Since the Gaussian distribution is a $C^\infty$ function, $P_R(r)$ is thus not only $C^1$ but also $C^\infty$. When $\Phi(H) + \sigma\epsilon$ is applied, the convolution with Gaussian noise significantly enhances the smoothness of the distribution, eliminating sharp variations and discontinuities present in the original distribution.

**Proposition 4.2** (Equivalence of CGFM and Learning Residual Distribution via Flow Matching). *Under Proposition 4.1 (noise smoothing) and two-sided coupling $\pi_{0,1|H} = p(x_0|h)q(x_1|h)$, CGFM equivalently learns residual $\epsilon = x_1 - x_0$'s probabilistic characteristics via Flow Matching, with its path and loss tied to $\epsilon$'s evolution. The proof is given in Appendix A.5.*

This equivalence highlights a critical advantage of CGFM: unlike discriminative models that directly output residuals to add back to predictions, CGFM explicitly models the target distribution (of future

sequences) by treating the auxiliary model's predictions as the initial distribution. By learning the transformation path from this initial prediction distribution to the target distribution via flow matching, it naturally captures the full structure of residuals—all without explicitly outputting residuals. Instead, enhanced forecasts are obtained by sampling directly from the learned target distribution.

**A Discussion on Residual Learning Between CGFM and Discriminative Models** We conducted a theoretical analysis to further underscore CGFM's superiority over discriminative models for residual learning. As detailed in the proof in Appendix A.2, discriminative models— which minimize mean squared error to learn deterministic mappings from auxiliary outputs $X_0$ to residuals—suffer from three inherent limitations. First, their loss function decomposes to only optimize the residual's conditional mean (first-order moment), inherently discarding critical higher-order distributional structures such as variance and skewness. Second, their input is restricted to $X_0$, rendering them unable to integrate historical context $H$ and thus incapable of capturing context-dependent residual dynamics tied to temporal trends. Third, their point-estimation nature imposes an irreducible risk lower bound (equal to the average conditional variance of residuals), which cannot be eliminated via parameter optimization.

In contrast, CGFM overcomes these flaws by explicitly learning the full probabilistic characteristics of residuals (Proposition 4.2), integrating historical context $H$ through its two-sided conditional design, and modeling the velocity field that governs residual evolution. This comprehensive framework enables CGFM to capture both the distributional patterns of residuals and their temporal dependencies, thereby explaining its consistent performance superiority.

## 4.4 VELOCITY FIELD OF MARGINAL PROBABILITY PATHS

Previous studies have primarily focused on the application of conditional optimal transport flows (Kollovieh et al., 2025; Hu et al., 2025). In the scenario of a two-sided condition, this can be formulated as: $X_t \sim p_{t|0,1,H} = tX_1 + (1 - t)X_0$. Conditional optimal transport flows addresses the problem of kinetic energy minimization through the optimization formulation: $\arg \min_{p_t, u_t} \int_0^1 \int_Z \|u_t(x)\|^2 p_t(x) \, dx \, dt$, providing a principled approach to solving such problems. However, in time-series forecasting, this may not identify the optimal predictive path, particularly when the initial distribution is highly complex. Notably, conditional optimal transport can be regarded as a special case within the broader family of affine conditional flows (Albergo & Vanden-Eijnden, 2023).

$$X_t \sim p_{t|0,1,H} = \alpha_t X_1 + \beta_t X_0, \tag{4}$$

where $\alpha_t$ and $\beta_t : [0, 1] \to [0, 1]$ are smooth functions, satisfying $\alpha_0 = \beta_1 = 0$ and $\alpha_1 = \beta_0 = 1$, with $\dot{\alpha}_t > 0$, and $-\dot{\beta}'_t > 0$ for $t \in (0, 1)$. Referring to Eq.(1), let $x' = \psi_t(x)$, and the inverse function yields $\psi_t^{-1}(x') = x$. Consequently, Eq.(1) can be reformulated as:

$$u_t(x') = \dot{\psi}_t(\psi_t^{-1}(x')). \tag{5}$$

**Lemma 4.3** (Velocity Field of Marginal Probability Paths). *Under mild assumptions, if $\psi_t(\cdot|x_0, x_1, h)$ is smooth in $t$ and forms a diffeomorphism in $x_0, x_1$, then the velocity field $u_t(x)$ can be represented as*

$$u_t(x|h) = \mathbb{E}\left[\dot{\psi}_t(X_0, X_1|H)|X_t = x, H = h\right], \tag{6}$$

*for all $t \in [0, 1)$. The proof is given in Appendix A.6.*

According to Lemma 4.3, under the two-sided condition, the velocity field takes the form of:

$$u_t(x|h) = \mathbb{E}\left[\dot{\alpha}_t X_1 + \dot{\beta}_t X_0 \mid X_t = x, H = h\right]. \tag{7}$$

The choice of $\alpha_t$ and $\beta_t$ enhances flexibility, making it better suited for complex source distributions and more effective for predictive path construction. We further investigate the effects of different parameterizations of $\alpha_t$ and $\beta_t$ in Experiment 5.6.

## 4.5 NON-CROSSING OF PROBABILITY PATHS

After deriving the conditional velocity field $u_t(x|h)$ and the conditional probability path $p_{t|H}(x|h)$ from the marginal velocity field, we must also ensure that $u_t(x|h)$ indeed generates $p_{t|H}(x|h)$.

**Lemma 4.4.** *Under mild assumptions, $q$ has a bounded support and $p$ is $C^1(\mathbb{R}^d)$ with a strictly positive density and finite second moments. These two are related by the conditional independent coupling $\pi_{0,1|H}(x_0, x_1|h) = p(x_0|h)q(x_1|h)$. $p_t(x|h)$ is defined as Eq.(4.2), with $\psi_t$ defined by Eq.(4). Subsequently, the marginal velocity engenders $p_t$ that interpolates between $p$ and $q$. The proof is given in Appendix A.7.*

According to Proposition 4.1, whether $p$ represents the noise distribution or the output distribution of the auxiliary model, appropriate operations can ensure that the $C^1$ condition is satisfied. Consequently, Proposition 4.4 guarantees the correctness of the flow matching construction.

**Proposition 4.5** (Transportation and Non-crossing of Probability Paths). *Under the assumptions of Proposition 4.4, further suppose that the affine conditional path and the velocity field are given by Eq. (4) and Eq. (7), respectively. Then All paths of the flow are non-crossing: there exist no $t \in [0,1)$, $z \in \mathbb{R}^d$, and distinct initial conditions $X_0^{(1)} \neq X_0^{(2)}$ such that $X_t^{(1)} = X_t^{(2)} = z$ with distinct evolution directions. The proof is given in Appendix A.8.*

As visualized in Figure 3, this guarantee of non-crossing paths is critical for time series forecasting, as path crossing would lead to ambiguous mappings between the auxiliary model's predictions $x_0$ and the true future values $x_1$ at intermediate time steps $t$. Such ambiguity could cause information loss or conflicting signals during the flow's evolution, undermining the model's ability to refine predictions consistently. For further discussion and detailed elaboration, refer to Appendix A.9.

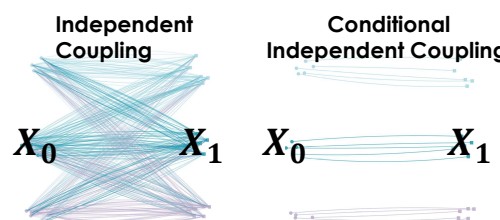

Figure 3: Illustration of CGFM's path properties. Left: Independent coupling may produce crossing paths. Right: CGFM's conditional independent coupling ensures non-crossing paths by conditioning both source and target on shared history.

### 4.6 PARAMETERIZATION OF THE PREDICTION TARGET

Numerous studies in the domains of time-series forecasting and protein synthesis(Watson et al., 2023; Shen et al., 2024; Shen & Kwok, 2023a), have undertaken the reparameterization of prediction targets. In the case of our two-sided condition's affine path By using Eq.(4), we obtain: $X_1 = \frac{X_t - \beta_t X_0}{\alpha_t}, X_0 = \frac{X_t - \alpha_t X_1}{\beta_t}$. Substituting it into Eq.(7) allows for the following reparameterization:

$$u_t(x|h) = \dot{\alpha}_t E[X_1|X_t = x, h] + \dot{\beta}_t E[X_0|X_t = x, h] \tag{8}$$

$$= \frac{\dot{\beta}_t}{\beta_t}x + \left[\dot{\alpha}_t - \frac{\alpha_t \dot{\beta}_t}{\beta_t}\right] E[X_1|X_t = x, H = h] \tag{9}$$

$$= \frac{\dot{\alpha}_t}{\alpha_t}x + \left[\dot{\beta}_t - \frac{\beta_t \dot{\alpha}_t}{\alpha_t}\right] E[X_0|X_t = x, H = h]. \tag{10}$$

Whereas Eq.(9) provides a parameterization of $u_t$ for predicting the target $x_1$, where $x_{1|t}(x) = \mathbb{E}[X_1|X_t = x]$ is defined as the $x_1$-prediction. Eq.(10) offers a parameterization of $u_t$ for the source $x_0$, where $x_{0|t}(x) = \mathbb{E}[X_0|X_t = x]$ is defined as the $x_0$-prediction. These equations introduce two novel methods of parameterization. In light of pertinent literature (Watson et al., 2023) and experimental observations, we discern that for sequence prediction tasks, considering our generation target $x_1$ as our training objective engenders enhanced outcomes.

### 4.7 LOSS FUNCTION

After obtaining the conditional guided probability path $p_{t|H}(x|h)$ and the velocity field $u_t(x|h)$, we proceed to define the loss function, specifically the guided flow matching loss:

$$\mathcal{L}_{GM}(\theta) = \mathbb{E}_{t,H,X_t}\left[\left\|u_t(X_t \mid H) - u_t^\theta(X_t \mid H)\right\|^2\right], \tag{11}$$

where the velocity field $u_t(X_t \mid H)$ is defined by: $u_t(X_t \mid H) = \mathbb{E}\left[ g_t(X_0, X_1) \,\Big|\, X_t = x, H = h \right]$.

The prediction function $g_t(X_0, X_1)$ is specified based on the prediction target as follows:

$$
g_t(X_0, X_1) = \begin{cases} \dot{\alpha}_t X_1 + \dot{\beta}_t X_0, & u_t\text{-Prediction}, \\ X_0, & X_0\text{-Prediction}, \\ X_1, & X_1\text{-Prediction}. \end{cases} \tag{12}
$$

From Eq.(8), Eq.(9), and Eq.(10), it can be shown that the above three prediction methods are mathematically equivalent. If the prediction objective is $X_1$-prediction, then after training the velocity field $u_t^\theta$ to predict $X_1$, it can be substituted into Eq. 9 to replace $\mathbb{E}[X_1 \mid X_t = x, H = h]$, resulting in the velocity field at time $t$. The same principle applies to $x_0$. Furthermore, the conditional guided flow matching loss function $\mathcal{L}_{CGM}(\theta)$ is defined as:

$$
\mathcal{L}_{CGM}(\theta) = \mathbb{E}_{t, H, (X_0, X_1) \sim \pi_{0,1 \mid H}}\left[ \left\| g_t(X_0, X_1) - u_t^\theta(X_t) \right\|^2 \right]. \tag{13}
$$

Since $g_t(X_0, X_1)$ is explicitly specified and computable, the loss $\mathcal{L}_{CGM}(\theta)$ offers significant advantages for optimization.

**Proposition 4.6.** *The gradients of the guided flow matching loss and the conditional guided flow matching loss coincide:*

$$
\nabla_\theta \mathcal{L}_{GM}(\theta) = \nabla_\theta \mathcal{L}_{CGM}(\theta). \tag{14}
$$

*Moreover, the minimizer of the Conditional Guided Flow Matching loss $\mathcal{L}_{CGM}(\theta)$ is the marginal velocity $u_t(X_t \mid H)$. The proof is given in Appendix A.9.*

## 5 EXPERIMENT

### 5.1 BASELINE AND DATASETS

To demonstrate the forecast enhancement effect and predictive superiority of CGFM, we employed representative baselines. Among transformer-based models, we included Multipatchformer (Naghashi et al., 2025), iTransformer (Liu et al., 2024), PatchTST (Nie et al., 2023), Pathformer (Chen et al., 2024), Autoformer (Chen et al., 2021), and the classic FedFormer (Zhou et al., 2022). Additionally, we incorporated MLP-based models, including RLinear (Li et al., 2023b), TimesNet (Wu et al., 2022), Timemixer (Wang et al., 2024) and TiDE (Das et al., 2024). Furthermore, diffusion-based models such as CSDI (Tashiro et al., 2021) and TimeDiff (Shen & Kwok, 2023a) were also evaluated. Descriptions of the datasets and their statistical properties are detailed in the Appendix A.13.

### 5.2 EVALUATION METRICS

The experiments employed Mean Squared Error (MSE), Mean Absolute Error (MAE), and Continuous Ranked Probability Score (CRPS) to evaluate the predictive performance of the models. To ensure the robustness of the results, each experiment was repeated 10 times, and the outcomes were averaged.

### 5.3 AUXILIARY MODEL ENHANCED PERFORMANCE

As shown in Table 1, for an input length of 96, the auxiliary model is evaluated across a wide range of mainstream forecasting models, including MLP-based, Transformer-based, and diffusion-based architectures. Our proposed CGFM framework achieves significant performance improvements across most benchmark datasets. Specifically, CGFM yields the most significant improvement when using Rlinear as the auxiliary model, whereas gains for iTransformer and TimeDiff are comparatively modest. This phenomenon is detailed in Figure 4, where the first three left panels show the PCA of CGFM's initial distributions (prediction distributions of Rlinear, iTransformer, and TimeDiff, respectively), and the rightmost panel shows the PCA of the target (ground truth) distribution. The PCA trajectory of Rlinear predictions closely aligns with the ground truth in both shape and continuity, which facilitates CGFM's learning of the transition from Rlinear's prediction distribution to the ground truth—resulting in substantial forecasting gains. In contrast, iTransformer and TimeDiff predictions

Table 1: Forecasting errors under the multivariate setting. The **bold** values indicate better performance. Note: Ex. = Exchange, Weath. = Weather. Comparison of four baseline models (Rlinear (2023), iTransformer (2024), TimeDiff (2023), MultiPatchFormer (2025)

| Methods Metric | Rlinear MSE | MAE | + CGFM MSE | MAE | iTrans. MSE | MAE | + CGFM MSE | MAE | TimeDiff MSE | MAE | + CGFM MSE | MAE | Multi. MSE | MAE | + CGFM MSE | MAE |
|---|---|---|---|---|---|---|---|---|---|---|---|---|---|---|---|---|
| **ETTm1** 96 | 0.359 | 0.378 | **0.307** | **0.351** | 0.336 | 0.369 | **0.313** | **0.362** | 0.339 | 0.362 | **0.309** | **0.361** | 0.317 | 0.345 | **0.308** | **0.355** |
| 192 | 0.396 | 0.395 | **0.341** | **0.382** | 0.387 | 0.392 | **0.366** | **0.382** | 0.372 | 0.381 | **0.346** | **0.389** | 0.367 | 0.369 | **0.339** | **0.378** |
| 336 | 0.428 | 0.416 | **0.372** | **0.397** | 0.427 | 0.422 | **0.398** | **0.412** | 0.403 | 0.401 | **0.384** | **0.409** | 0.399 | 0.398 | **0.373** | **0.401** |
| 720 | 0.489 | 0.451 | **0.443** | **0.421** | 0.493 | 0.461 | **0.461** | **0.452** | 0.455 | 0.432 | **0.441** | **0.416** | 0.467 | 0.436 | **0.438** | **0.430** |
| **ETTm2** 96 | 0.182 | 0.267 | **0.167** | **0.253** | 0.179 | 0.262 | **0.177** | **0.259** | 0.185 | 0.265 | **0.170** | **0.261** | 0.171 | 0.252 | **0.165** | **0.255** |
| 192 | 0.246 | 0.305 | **0.228** | **0.298** | 0.244 | 0.306 | **0.242** | **0.299** | 0.251 | 0.310 | **0.234** | **0.286** | 0.238 | 0.296 | **0.229** | **0.296** |
| 336 | 0.310 | 0.344 | **0.281** | **0.323** | 0.314 | 0.351 | **0.291** | **0.329** | 0.311 | 0.352 | **0.283** | **0.315** | 0.305 | 0.342 | **0.283** | **0.320** |
| 720 | 0.407 | 0.399 | **0.365** | **0.367** | 0.413 | 0.407 | **0.380** | **0.391** | 0.412 | 0.399 | **0.373** | **0.386** | 0.404 | 0.403 | **0.366** | **0.371** |
| **ETTh1** 96 | 0.382 | 0.398 | **0.363** | **0.372** | 0.389 | 0.408 | **0.368** | **0.388** | 0.383 | 0.391 | **0.371** | **0.386** | 0.378 | 0.389 | **0.365** | **0.371** |
| 192 | 0.439 | 0.424 | **0.409** | **0.417** | 0.443 | 0.441 | **0.410** | **0.423** | 0.437 | 0.429 | **0.415** | **0.421** | 0.434 | 0.422 | **0.422** | **0.435** |
| 336 | 0.480 | 0.448 | **0.425** | **0.430** | 0.489 | 0.461 | **0.428** | **0.437** | 0.475 | 0.449 | **0.423** | **0.432** | 0.473 | 0.445 | **0.439** | **0.48** |
| 720 | 0.484 | 0.475 | **0.461** | **0.457** | 0.506 | 0.498 | **0.503** | **0.497** | 0.502 | 0.512 | **0.476** | **0.495** | 0.476 | 0.470 | **0.462** | **0.461** |
| **ETTh2** 96 | 0.290 | 0.341 | **0.275** | **0.329** | 0.299 | 0.351 | **0.295** | **0.348** | 0.301 | 0.357 | **0.282** | **0.346** | 0.285 | 0.334 | **0.280** | **0.328** |
| 192 | 0.375 | 0.392 | **0.351** | **0.362** | 0.383 | 0.402 | **0.377** | **0.398** | 0.381 | 0.396 | **0.372** | **0.391** | 0.371 | 0.389 | **0.365** | **0.372** |
| 336 | 0.414 | 0.426 | **0.402** | **0.422** | 0.431 | 0.435 | **0.423** | **0.431** | 0.433 | 0.441 | **0.425** | **0.432** | 0.420 | 0.428 | **0.415** | **0.417** |
| 720 | 0.422 | 0.447 | **0.411** | **0.442** | 0.429 | 0.448 | **0.423** | **0.445** | 0.437 | 0.458 | **0.419** | **0.445** | 0.425 | 0.441 | **0.419** | **0.432** |
| **Ex.** 96 | 0.095 | 0.215 | **0.081** | **0.204** | 0.089 | 0.218 | **0.082** | **0.206** | 0.087 | 0.212 | **0.082** | **0.203** | 0.085 | 0.206 | **0.080** | **0.201** |
| 192 | 0.182 | 0.308 | **0.175** | **0.304** | 0.177 | 0.301 | **0.174** | **0.308** | 0.176 | 0.311 | **0.174** | **0.307** | 0.178 | 0.297 | **0.173** | **0.302** |
| 336 | 0.349 | 0.432 | **0.306** | **0.395** | 0.336 | 0.421 | **0.306** | **0.397** | 0.310 | 0.427 | **0.305** | **0.399** | 0.307 | 0.399 | **0.304** | **0.394** |
| 720 | 0.890 | 0.719 | **0.830** | **0.683** | 0.851 | 0.693 | **0.837** | **0.690** | 0.847 | 0.706 | **0.844** | **0.701** | 0.897 | 0.702 | **0.844** | **0.685** |
| **Traffic** 96 | 0.632 | 0.387 | **0.412** | **0.288** | 0.397 | 0.272 | **0.388** | **0.268** | 0.520 | 0.373 | **0.398** | **0.277** | 0.442 | 0.268 | **0.423** | **0.253** |
| 192 | 0.597 | 0.362 | **0.429** | **0.291** | 0.422 | 0.278 | **0.413** | **0.269** | 0.515 | 0.354 | **0.427** | **0.281** | 0.460 | 0.273 | **0.441** | **0.266** |
| 336 | 0.607 | 0.369 | **0.462** | **0.336** | 0.437 | 0.288 | **0.428** | **0.276** | 0.514 | 0.355 | **0.459** | **0.322** | 0.477 | 0.276 | **0.454** | **0.271** |
| 720 | 0.650 | 0.391 | **0.489** | **0.323** | 0.473 | 0.304 | **0.462** | **0.296** | 0.563 | 0.377 | **0.478** | **0.310** | 0.517 | 0.299 | **0.471** | **0.286** |
| **Weath.** 96 | 0.189 | 0.230 | **0.152** | **0.191** | 0.178 | 0.217 | **0.154** | **0.193** | 0.181 | 0.217 | **0.156** | **0.191** | 0.157 | 0.197 | **0.151** | **0.186** |
| 192 | 0.244 | 0.275 | **0.201** | **0.226** | 0.224 | 0.259 | **0.204** | **0.241** | 0.228 | 0.257 | **0.202** | **0.239** | 0.207 | 0.242 | **0.195** | **0.221** |
| 336 | 0.295 | 0.309 | **0.259** | **0.272** | 0.281 | 0.298 | **0.270** | **0.289** | 0.288 | 0.303 | **0.273** | **0.292** | 0.277 | 0.293 | **0.243** | **0.254** |
| 720 | 0.368 | 0.355 | **0.339** | **0.332** | 0.359 | 0.350 | **0.343** | **0.334** | 0.364 | 0.358 | **0.347** | **0.340** | 0.351 | 0.342 | **0.332** | **0.321** |

Figure 4: PCA visualization of predictions and ground truth, showing the PCA projections of RLinear predictions, iTransformer predictions, TimeDiff predictions, and the ground truth, respectively.

exhibit irregular spatial distributions with discontinuous inter-point connections, indicating non-smooth fluctuations in their high-dimensional representations. This increases CGFM's learning difficulty, leading to less pronounced improvements. Detailed procedures are provided in Appendix A.7.2. Further results regarding CRPS are provided in Appendix A.10.

## 5.4 PERFORMANCE WITHOUT AUXILIARY MODEL

Table 2: Testing MSE in the multivariate setting. Number in brackets is the rank. CSDI runs out of memory on *Traffic*.

| Model | Weather | Traffic | ETTh1 | ETTh2 | ETTm1 | ETTm2 | Exchange | Avg Rank |
|---|---|---|---|---|---|---|---|---|
| **CGFM** | 0.161(3) | 0.430(2) | 0.373(1) | 0.286(2) | 0.317(2) | 0.173(2) | 0.085(3) | **2.142 (1)** |
| **TimeDiff** | 0.181(8) | 0.520(7) | 0.383(7) | 0.301(7) | 0.339(8) | 0.185(8) | 0.087(6) | 7.125(7) |
| **CSDI** | 0.301(13) | - | 0.503(13) | 0.356(11) | 0.601(13) | 0.289(13) | 0.258(13) | 12.667(13) |
| **iTransformer** | 0.178(6) | 0.397(1) | 0.389(10) | 0.299(6) | 0.336(6) | 0.179(6) | 0.089(7) | 6.000(6) |
| **Rlinear** | 0.189(9) | 0.632(11) | 0.382(5) | 0.290(4) | 0.359(9) | 0.182(7) | 0.095(8) | 7.429(8) |
| **FedFormer** | 0.219(11) | 0.588(8) | 0.376(3) | 0.359(12) | 0.379(11) | 0.203(10) | 0.147(11) | 7.571(9) |
| **TimeMixer** | 0.165(4) | 0.461(4) | 0.374(2) | 0.294(5) | 0.331(5) | 0.175(4) | 0.083(1) | 3.571(4) |
| **TimesNet** | 0.179(7) | 0.593(9) | 0.384(9) | 0.340(9) | 0.338(7) | 0.188(9) | 0.107(10) | 8.571(10) |
| **PatchTST** | 0.177(5) | 0.462(5) | 0.383(7) | 0.304(8) | 0.320(4) | 0.175(4) | 0.085(3) | 5.142(5) |
| **Autoformer** | 0.266(12) | 0.613(10) | 0.449(11) | 0.345(10) | 0.505(12) | 0.255(12) | 0.189(12) | 11.286(12) |
| **TiDE** | 0.202(10) | 0.803(12) | 0.478(12) | 0.403(13) | 0.366(10) | 0.209(11) | 0.093(8) | 10.857(11) |
| **Pathformer** | 0.156(1) | 0.479(6) | 0.382(5) | 0.283(1) | 0.319(3) | 0.174(3) | 0.083(1) | 2.857(3) |
| **MultiPatchFormer** | 0.158(2) | 0.438(2) | 0.378(4) | 0.285(3) | 0.315(1) | 0.171(1) | 0.085(3) | 2.285(2) |

To further validate the superiority of our proposed forecasting framework, we conducted an additional experiment where the initial condition $x_0$ was directly set as noise under the 96-to-96 prediction

Table 3: MSE and MAE results for different affine conditional paths on the ETTh1 and ETTm1 datasets.

| Dataset | CondOT | | Poly-n | | VP | | Cosine | |
|---|---|---|---|---|---|---|---|---|
| | MSE | MAE | MSE | MAE | MSE | MAE | MSE | MAE |
| ETTh1 (Rlinear) | 0.368 | 0.376 | **0.363** | **0.372** | 0.379 | 0.387 | 0.380 | 0.396 |
| ETTm1 (Rlinear) | 0.314 | 0.363 | **0.307** | **0.351** | 0.336 | 0.376 | 0.332 | 0.371 |
| ETTh1 (iTrans) | 0.374 | 0.391 | **0.368** | **0.388** | 0.387 | 0.403 | 0.387 | 0.386 |
| ETTm1 (iTrans) | 0.326 | 0.367 | **0.313** | **0.362** | 0.331 | 0.372 | 0.334 | 0.376 |

Table 4: MSE and MAE results for different prediction functions on ETTh1 and ETTm1 datasets.

| Dataset | $u_t$-Prediction | | $X_0$-Prediction | | $X_1$-Prediction | |
|---|---|---|---|---|---|---|
| | MSE | MAE | MSE | MAE | MSE | MAE |
| ETTh1 | 0.370 | 0.379 | 0.384 | 0.385 | **0.363** | **0.372** |
| ETTm1 | 0.328 | 0.361 | 0.343 | 0.367 | **0.307** | **0.351** |

setting. CGFM achieved the best overall ranking, demonstrating the effectiveness and soundness of its model architecture. Although it does not attain the best performance on every individual dataset, as shown in Table 1, when combined with an auxiliary model, CGFM consistently achieves the best results across all datasets. This highlights the crucial role of the CGFM auxiliary model in enhancing forecasting performance.

### 5.5 CASE STUDY OF RESIDUAL LEARNING

As shown in Figure 5 and Figure 6, we evaluate the 96-to-96 prediction task on the ETTh1 dataset using RLinear as the auxiliary model. The curves represent the mean sequences obtained by averaging all predicted and ground truth windows of length 96. It can be observed that RLinear exhibits persistent underestimation and large variance. In contrast, CGFM effectively learns the residual distribution, substantially reducing both variance and systematic bias, resulting in predictions that closely align with the ground truth. Detailed procedures are provided in Appendix A.14.1.

### 5.6 ANALYSIS OF PATH HYPERPARAMETER

We explore different parameterization schemes for affine probabilistic paths, including Optimal Transport (CondOT), Polynomial (Poly-n), Linear Variance Preserving (LinearVP), and Cosine schedulers, with their comparisons provided in Figure 7. Detailed formulas are provided in Appendix A.5.

Table 3 shows that the Poly-n scheduler, with velocity approaching zero near $t \approx 0$, enables thorough exploration around $X_0$, similar to the denoising phase in diffusion models. This extension effectively increases the model's decision time, enhancing its ability to capture intricate details early on.

### 5.7 ANALYSIS OF PREDICTION FUNCTIONS

To study different prediction functions' impact on time-series forecasting, we tested ETTh1 and ETTm1 datasets with RLinear as the auxiliary model. Though mathematically equivalent, results (Table 4) show $X_1$-Prediction consistently achieves the lowest MSE and MAE, outperforming $X_0$- and $u_t$-Prediction.

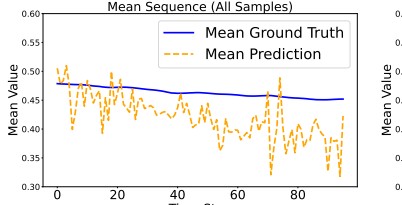 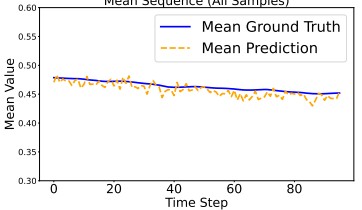

Figure 5: Rlinear Without CGFM    Figure 6: Rlinear With CGFM

Intuitively, $X_1$-Prediction directly targets the future series, $X_0$-Prediction focuses on noise, and $u_t$-Prediction mixes both. While the equivalence is valid only under the theoretical optimum, in practical training $X_1$-Prediction generally exhibits faster convergence and greater ease of learning. This aligns with prior work (Watson et al., 2023; Shen & Kwok, 2023a).

## 6 CONCLUSION

In this paper, we propose Conditional Guided Flow Matching (CGFM), a novel framework that advances time series forecasting by explicitly modeling the full structure of prediction residuals. Diverging from treating residuals as mere optimization targets, CGFM leverages flow matching's flexibility, using an auxiliary model's output as the source distribution. We tailor the framework with integrated innovations: an innovative two-sided conditional probability path, general affine paths for flow construction, noise smoothing for robustness, and a velocity field adapted with specialized prediction target parameterization for time series enhancement. This design is theoretically critical—it avoids path crossing, preserves temporal consistency, and is proven superior to discriminative residual learning models. Extensive experiments confirm CGFM's effectiveness and generality: it improves forecasting across datasets, acting as a robust refinement tool for time series forecasting.

## 7 ETHICS STATEMENT

This work adheres to the ICLR Code of Ethics. We use publicly available time series datasets (e.g., Weather, Traffic, ETT, Exchange) in strict compliance with their licensing agreements, ensuring no personal/sensitive information is involved and maintaining data provenance transparency. We uphold scientific integrity by accurately reporting experimental results, providing detailed reproducible model implementations, and properly citing prior work. This research complies with institutional ethics guidelines and core principles of responsible, transparent, and non-harmful research practice.

## 8 REPRODUCIBILITY STATEMENT

To ensure reproducibility, we provide anonymous source code for the CGFM framework as supplementary material. Detailed experimental settings, Data preprocessing steps for all public datasets and complete proofs of theoretical propositions are included in the supplementary materials and Appendix, respectively. These resources enable reproduction of our results and verification of claims.

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

# A APPENDIX

## A.1 PROOFS OF THE PROPOSITIONS

**Lemma A.1** (Mass Conservation and Flow Generation). *Let $p_t(x)$ be a probability density on $\mathbb{R}^d$ for $t \in [0, 1)$, and $u_t(x)$ a vector field satisfying: for all $(t, x) \in [0, 1) \times \mathbb{R}^d$, there exists a neighborhood $\mathcal{U} \subset \mathbb{R}^d$ where $u_t$ is Lipschitz in $x$. $\int_0^1 \int_{\mathbb{R}^d} \|u_t(x)\| p_t(x) dx dt < \infty$.*

*1. The pair $(u_t, p_t)$ satisfies the continuity equation:*

$$\frac{\partial}{\partial t} p_t(x) + \nabla_x \cdot \big(u_t(x)\, p_t(x)\big) = 0. \tag{15}$$

*2. The vector field $u_t$ generates $p_t$ through the flow map $\psi_t$, defined by:*

$$u_t \text{ generates } p_t \text{ if } X_t = \psi_t(X_0) \sim p_t \text{ for all } t \in [0, 1). \tag{16}$$

*Furthermore, the flow map satisfies the integral representation:*

$$\psi_t(x) = x + \int_0^t u_s(\psi_s(x)) ds, \tag{17}$$

*This lemma is built upon the fundamental framework of flow matching (Lipman et al., 2023), integrating the core properties of the continuity equation and flow mapping.*

**Proposition A.2** ($L^2$ Distance for Learning Conditional Expectations). *Let $X \in \mathbb{S}_X$, $Y \in \mathbb{S}_Y$ be random variables over state spaces $\mathbb{S}_X, \mathbb{S}_Y$, and let $g : \mathbb{R}^p \times \mathbb{S}_X \to \mathbb{R}^n$ be a function $(\theta, x) \to g^\theta(x)$, where $\theta \in \mathbb{R}^p$ denotes learnable parameters. Let $\|u - v\|^2$ denote the squared $L^2$ distance. Then:*

$$\nabla_\theta \mathbb{E}_{X,Y} \|Y - g^\theta(X)\|^2 = \nabla_\theta \mathbb{E}_X \|\mathbb{E}[Y|X] - g^\theta(X)\|^2. \tag{18}$$

*In particular, for all $x \in \mathbb{S}_X$ with $p_X(x) > 0$, the global minimum of $g^\theta(x)$ with respect to $\theta$ satisfies:*

$$g^\theta(x) = \mathbb{E}[Y \mid X = x]. \tag{19}$$

*Proof.* Assume $g^\theta$ is differentiable in $\theta$, and that differentiation and integration can be interchanged.

$$\begin{aligned}
\nabla_\theta \mathbb{E}_{X,Y} \|Y - g^\theta(X)\|^2 &= \mathbb{E}_{X,Y} \left[\nabla_\theta \|Y - g^\theta(X)\|^2\right] \\
&= \mathbb{E}_X \mathbb{E}_{Y|X} \left[2 \left(g^\theta(X) - Y\right)^\top \nabla_\theta g^\theta(X)\right] \\
&= \mathbb{E}_X \left[2 \left(g^\theta(X) - \mathbb{E}[Y|X]\right)^\top \nabla_\theta g^\theta(X)\right] \\
&= \nabla_\theta \mathbb{E}_X \|\mathbb{E}[Y|X] - g^\theta(X)\|^2
\end{aligned}$$

$\square$

This proposition is a direct generalization of the properties of least squares estimation in statistical learning

**Proposition A.3** (Noise Smoothing). *Let $X \in \mathbb{R}^{C \times F}$ be a time series with distribution $P_{ori}$. Define the perturbed series $R$ as*

$$R = X + \sigma \epsilon, \tag{20}$$

*where $\epsilon \sim \mathcal{N}(0, I)$ is additive Gaussian noise and $\sigma > 0$. Then, the perturbed series $R$ follows the distribution $P_{per}$:*

$$P_{per}(r) = \int P_X(r - \sigma \epsilon) P_\epsilon(\epsilon)\, d\epsilon, \tag{21}$$

*which belongs to $C^1(\mathbb{R}^d)$. Furthermore, $P_{per}$ has a strictly positive density and possesses finite second moments.*

*Proof.* Assume $X$ and $\epsilon$ are independent, and $P_X$ is a probability density function. By independence of $X$ and $\epsilon$, the density of $R = X + \sigma\epsilon$ is given by the convolution:

$$P_{\text{per}}(r) = \int P_X(r - \sigma\epsilon)P_\epsilon(\epsilon)\,d\epsilon. \tag{22}$$

The Gaussian density $P_\epsilon(\epsilon) = (2\pi)^{-d/2}e^{-\|\epsilon\|^2/2}$ is infinitely differentiable ($C^\infty$). Convolution with $P_X$ preserves smoothness. Specifically:

$$\nabla_r P_{\text{per}}(r) = \int \nabla_r P_X(r - \sigma\epsilon)P_\epsilon(\epsilon)\,d\epsilon. \tag{23}$$

Since $P_\epsilon \in C^\infty$ and decays exponentially, differentiation under the integral sign is justified by dominated convergence. Thus, $P_{\text{per}} \in C^1(\mathbb{R}^d)$. For any $r \in \mathbb{R}^d$, since $P_\epsilon(\epsilon) > 0$ everywhere and $\sigma > 0$, there exists $\epsilon$ such that $P_X(r - \sigma\epsilon) > 0$. Hence:

$$P_{\text{per}}(r) = \int P_X(r - \sigma\epsilon)P_\epsilon(\epsilon)d\epsilon > 0. \tag{24}$$

$$\mathbb{E}[\|X + \sigma\epsilon\|^2] = \mathbb{E}[\|X\|^2] + 2\sigma\mathbb{E}[X^\top\epsilon] + \sigma^2\mathbb{E}[\|\epsilon\|^2]. \tag{25}$$

By independence, $\mathbb{E}[X^\top\epsilon] = \mathbb{E}[X]^\top\mathbb{E}[\epsilon] = 0$. Since $\epsilon \sim \mathcal{N}(0, I)$, $\mathbb{E}[\|\epsilon\|^2] = d < \infty$. $X$ is a time series with $\mathbb{E}[\|X\|^2] < \infty$, all terms are finite, implying $\mathbb{E}[\|R\|^2] < \infty$. □

**Lemma A.4.** *Under the assumption that $p_t(x) > 0$ for all $(t, x)$, if $u_t(x|z)$ is conditionally integrable and generates the conditional probability path $p_t(\cdot|z)$, then the marginal velocity field $u_t(x)$ generates the marginal probability path $p_t(x)$ for all $t \in [0, 1)$.*

*Proof.* We verify the two conditions of the Mass Conservation Lemma through the following continuous derivation:

$$\begin{aligned}
\frac{\partial}{\partial t}p_t(x) &= \int \frac{\partial}{\partial t}p_{t|Z}(x|z)p_Z(z)dz \\
&= -\int \nabla_x \cdot \big[u_t(x|z)p_{t|Z}(x|z)\big]p_Z(z)dz \\
&= -\nabla_x \cdot \int u_t(x|z)p_{t|Z}(x|z)p_Z(z)dz \\
&= -\nabla_x \cdot [u_t(x)p_t(x)]
\end{aligned}$$

To verify integrability, apply vector Jensen's inequality to the conditional integrability condition:

$$\int_0^1 \int \|u_t(x)\|p_t(x)\,dxdt$$

$$\leq \int_0^1 \int \left(\int \|u_t(x|z)\|p_{t|Z}(x|z)p_Z(z)\,dz\right)dxdt$$

$$< \infty. \tag{26}$$

The Lipschitz continuity of $u_t(x)$ follows from the $C^1$-smoothness of $u_t(x|z)$ and $p_{t|Z}(x|z)$, which is preserved under convex combinations. By satisfying both the continuity equation and the integrability condition, the marginal velocity $u_t(x)$ generates $p_t(x)$ via the flow map $\psi_t(x) = x + \int_0^t u_s(\psi_s(x))ds$. □

This lemma is an extension of the flow matching framework (Lipman et al., 2023) to marginal distributions.

**Proposition A.5** (Equivalence of CGFM and Learning Residual Distribution via Flow Matching).
*Under Proposition A.3 (noise smoothing) and two-sided coupling $\pi_{0,1|H} = p(x_0|h)q(x_1|h)$, CGFM equivalently learns residual $\epsilon = x_1 - x_0$'s probabilistic characteristics via Flow Matching, with its path and loss tied to $\epsilon$'s evolution.*

*Proof.* Let the affine path be defined by the random variable $X_t = \alpha_t X_1 + \beta_t X_0$, with boundary conditions $\alpha_0 = 0, \beta_0 = 1$ and $\alpha_1 = 1, \beta_1 = 0$. Let the residual be the random variable $\epsilon = X_1 - X_0$.

The core of the CGFM framework is the probability path $p_t(x|h)$, which is the probability density function of the random variable $X_t$. The path interpolates between the source distribution $p(x_0|h)$ at $t = 0$ and the target distribution $q(x_1|h)$ at $t = 1$. We can express the random variable $X_t$ in terms of the random variables $X_0$ and $\epsilon$ by substituting $X_1 = X_0 + \epsilon$:

$$X_t = \alpha_t(X_0 + \epsilon) + \beta_t X_0 = (\alpha_t + \beta_t)X_0 + \alpha_t \epsilon \tag{27}$$

This equation shows that at any time $t$, the random variable $X_t$ is a linear transformation of the random variables $X_0$ and $\epsilon$. The distribution of $X_t$, which is $p_t(x|h)$, is therefore a direct consequence of the joint distribution of $(X_0, \epsilon)$ and this time-dependent transformation.

The initial distribution of the path, $p_0(x|h)$, corresponds to $X_0$, which is the auxiliary model's prediction. The final distribution, $p_1(x|h)$, corresponds to $X_1 = X_0 + \epsilon$, which is the true value. The entire sequence of distributions $p_t(x|h)$ for $t \in [0, 1]$ is a continuous evolution from the prediction's distribution to the true value's distribution. This evolution is driven by the transformation of the residual $\epsilon$. Therefore, learning the sequence of distributions $p_t(x|h)$ via Flow Matching is equivalent to learning how the distribution of the residual $\epsilon$ evolves to correct the initial prediction.

The CGFM loss function trains a neural network $u_t^\theta(x_t|h)$ to match the target velocity field $g_t(X_0, X_1) = \frac{d}{dt}X_t$. The velocity field represents the local probability flow that governs the evolution of the distribution $p_t(x|h)$. We express the velocity field in terms of the random variables $X_0$ and $\epsilon$:

$$g_t(X_0, X_1) = \dot{\alpha}_t X_1 + \dot{\beta}_t X_0 = (\dot{\alpha}_t + \dot{\beta}_t)X_0 + \dot{\alpha}_t \epsilon \tag{28}$$

The Flow Matching loss minimizes the difference between the model's output and this target field. By training the model to approximate $g_t$, we are forcing it to learn a function that predicts the instantaneous change in the random variables $X_0$ and $\epsilon$ that constitutes the probability flow. This process of learning the velocity field is equivalent to learning the probabilistic dynamics of the residual—that is, how its statistical properties (e.g., mean, variance) evolve over time to correct the initial prediction. $\qquad\square$

**Proposition A.6** (Velocity Field of Marginal Probability Paths). *Suppose the following conditions hold:* $\psi_t(\cdot|x_0, x_1, h)$ *is smooth in* $t$ *and forms a diffeomorphism over* $(x_0, x_1)$, $\pi_{0,1|H}(x_0, x_1|h)$ *is the joint distribution of* $(X_0, X_1)$ *given* $H = h$, $p_{t|H}(x|h) > 0$ *for all* $(t, x, h)$

*Then the velocity field of the conditional path is given by:*

$$u_t(x|h) = \mathbb{E}\left[\dot{\psi}_t(X_0, X_1|h) \mid X_t = x, H = h\right], \quad t \in [0, 1). \tag{29}$$

*Proof.* Assume the flow map $\psi_t$ is differentiable in $t$ and that differentiation and integration can be interchanged.

$$
\begin{aligned}
u_t(x \mid h) &= \int u_t(x \mid x_0, x_1, h)\, p_{0,1|t,H}(x_0, x_1 \mid x, h)\, dx_0\, dx_1 \\
&= \int \dot{\psi}_t(x_0, x_1 \mid h)\, \frac{p_{t|0,1,H}(x \mid x_0, x_1, h)}{p_{t|H}(x \mid h)} \\
&\quad \times \pi_{0,1|H}(x_0, x_1 \mid h)\, dx_0\, dx_1 \\
&= \mathbb{E}_{X_0, X_1 | X_t = x, H = h}\left[\dot{\psi}_t(X_0, X_1 \mid h)\right] \\
&= \mathbb{E}\left[\dot{\psi}_t(X_0, X_1 \mid h) \mid X_t = x, H = h\right].
\end{aligned}
$$

$\qquad\square$

**Proposition A.7** (Marginalization via Conditional Affine Flows). *Under mild regularity conditions, q have bounded support, and p be $C^1(\mathbb{R}^d)$ with strictly positive density and finite second moments. Given a conditional independent coupling $\pi_{0,1|H}(x_0, x_1|h) = p(x_0|h)q(x_1|h)$, define the marginal probability path via affine interpolation:*

$$p_t(x|h) = \int p_{t|0,1,H}(x|x_0, x_1, h)\pi_{0,1|H}(x_0, x_1|h)dx_0 dx_1, \tag{30}$$

*where $X_t = \alpha_t X_1 + \beta_t X_0$ with $\alpha_t, \beta_t \in C^1([0,1])$. Then the marginal velocity field $u_t(x)$ generates a probability path $p_t(x)$ interpolating p and q.*

*Proof.* By Lemma A.4, it suffices to verify:

1. Conditional integrability: $\int_0^1 \mathbb{E}[\|u_t(X_t|h)\|]dt < \infty$.

2. Boundary conditions: $p_0 = p$ and $p_1 = q$

For the affine flow $X_t = \alpha_t X_1 + \beta_t X_0$, the velocity field is:

$$u_t(X_t|h) = \dot{\alpha}_t X_1 + \dot{\beta}_t X_0$$

$$\mathbb{E}[\|u_t(X_t|h)\|] \leq |\dot{\alpha}_t|\mathbb{E}[\|X_1\|] + |\dot{\beta}_t|\mathbb{E}[\|X_0\|]$$

Since $q$ has bounded support, $\mathbb{E}[\|X_1\|] < C_q < \infty$. For $p$ with finite second moments:

$$\mathbb{E}[\|X_0\|] \leq \sqrt{\mathbb{E}[\|X_0\|^2]} < \infty$$

The time integrals satisfy:

$$\int_0^1 (|\dot{\alpha}_t|C_q + |\dot{\beta}_t|C_p)dt \leq C(\|\dot{\alpha}\|_{L^1} + \|\dot{\beta}\|_{L^1}) < \infty$$

At endpoints:

$$t = 0: \quad X_0 = 0 \cdot X_1 + 1 \cdot X_0 \sim p(\cdot|h)$$
$$t = 1: \quad X_1 = 1 \cdot X_1 + 0 \cdot X_0 \sim q(\cdot|h)$$

Marginalizing over $h$ preserves the boundary conditions. $\square$

**Proposition A.8** (Transportation and Non-crossing of Probability Paths). *Under the assumptions of Proposition 4.4, further suppose that the affine conditional path is given by $X_t = \alpha_t X_1 + \beta_t X_0$ with $\alpha_t, \beta_t \in C^1([0,1])$, $\dot{\alpha}_t > 0$, and $-\dot{\beta}_t > 0$ for $t \in (0,1)$. The velocity field is defined as $u_t(x \mid h) = \mathbb{E}\left[\dot{\alpha}_t X_1 + \dot{\beta}_t X_0 \mid X_t = x, H = h\right]$. Then all paths of the flow are non-crossing: there exist no $t \in [0,1)$, $z \in \mathbb{R}^d$, and distinct initial conditions $X_0^{(1)} \neq X_0^{(2)}$ such that $X_t^{(1)} = X_t^{(2)} = z$.*

*Proof.* To establish the non-crossing property, we rely on the uniqueness of solutions to the probability flow ODE:

$$\frac{dX_t}{dt} = u_t(X_t \mid h). \tag{31}$$

The existence and uniqueness of solutions for an ODE are guaranteed by the Picard-Lindelöf theorem if the velocity field $u_t(x \mid h)$ is locally Lipschitz continuous with respect to the spatial variable $x$.

From the assumptions, we have the following:

1. The source distribution $p(x_0 \mid h)$ is $C^1(\mathbb{R}^d)$ with a strictly positive density.

2. The affine path parameters $\alpha_t, \beta_t \in C^1([0,1])$ with $\dot{\alpha}_t > 0$ and $-\dot{\beta}_t > 0$.

These conditions ensure that the velocity field $u_t(x \mid h)$, being a conditional expectation of a smooth function of $(X_0, X_1)$ given the smooth transformation $X_t = x$, is locally Lipschitz continuous in $x$ for each $t \in [0,1)$. This is a standard result in the theory of conditional expectations and smooth transformations.

Now, we proceed with a proof by contradiction. Suppose there exist two distinct initial conditions $X_0^{(1)} \neq X_0^{(2)}$ and a time $t^* \in (0, 1)$ such that their corresponding solution paths intersect at a point $z \in \mathbb{R}^d$. That is, $X_{t^*}^{(1)} = X_{t^*}^{(2)} = z$.

At time $t^*$, we have two solutions to the ODE equation 31 that both pass through the same point $z$. However, the uniqueness of solutions guaranteed by the Picard-Lindelöf theorem implies that if two solution paths coincide at a single point in time, they must be identical for all subsequent times $s \geq t^*$. Moreover, since the affine transformation is invertible for $t \in (0, 1)$, the uniqueness extends backwards in time as well. Therefore, the two paths must be identical for all $s \in [0, 1)$, which implies that their initial conditions must also be identical: $X_0^{(1)} = X_0^{(2)}$.

This contradicts our initial assumption that $X_0^{(1)} \neq X_0^{(2)}$. Therefore, no two distinct paths of the flow can cross. $\qquad\square$

**Proposition A.9** (Gradient Equivalence). *Let $t \in [0, 1]$, $H \sim p_H$, $(X_0, X_1) \sim \pi_{0,1|H}$, and $X_t$ be generated by a bridge process conditioned on $(X_0, X_1, H)$. Define: $\mathcal{L}_{GM}(\theta) = \mathbb{E}_{t,H,X_t} \|u_t(X_t|H) - u_t^\theta(X_t|H)\|^2$ , $\mathcal{L}_{CGM}(\theta) = \mathbb{E}_{t,H,X_0,X_1} \|g_t(X_0, X_1) - u_t^\theta(X_t|H)\|^2$*

*where $u_t(x|h) = \mathbb{E}[g_t(X_0, X_1)|X_t = x, H = h]$. Then:*

$$\nabla_\theta \mathcal{L}_{GM}(\theta) = \nabla_\theta \mathcal{L}_{CGM}(\theta) \tag{32}$$

*The minimizer of $\mathcal{L}_{CGM}(\theta)$ satisfies:*

$$u_t^\theta(x|h) = u_t(x|h), \quad \forall x, h, t \text{ with } p_t(x|h) > 0 \tag{33}$$

*Proof.* Assume $u_t^\theta$ is differentiable in $\theta$, and differentiation commutes with integration. Fix $t \in [0, 1]$ and $H = h$, define:

$$X \triangleq X_t|H = h \sim p_t(\cdot|h). \tag{34}$$
$$Y \triangleq g_t(X_0, X_1)|H = h \text{ with } (X_0, X_1) \sim \pi_{0,1|h}. \tag{35}$$

By Proposition A.2 applied to $(X, Y)$, we have:

$$\begin{aligned}
\nabla_\theta \mathbb{E}_{X_0, X_1} \|g_t - u_t^\theta(X_t|h)\|^2 &= \nabla_\theta \mathbb{E}_{X,Y} \|Y - u_t^\theta(X|h)\|^2 \\
&= \nabla_\theta \mathbb{E}_X \|\mathbb{E}[Y|X] - u_t^\theta(X|h)\|^2 \\
&= \nabla_\theta \mathbb{E}_{X_t} \|u_t(X_t|h) - u_t^\theta(X_t|h)\|^2
\end{aligned}$$

Integrate over $t \sim U[0, 1]$ and $H \sim p_H$:

$$\begin{aligned}
\nabla_\theta \mathcal{L}_{CGM}(\theta) &= \mathbb{E}_{t,H} \left[ \nabla_\theta \mathbb{E}_{X_0, X_1} \|g_t - u_t^\theta\|^2 \right] \\
&= \mathbb{E}_{t,H} \left[ \nabla_\theta \mathbb{E}_{X_t} \|u_t - u_t^\theta\|^2 \right] \\
&= \nabla_\theta \mathbb{E}_{t,H,X_t} \|u_t - u_t^\theta\|^2 \\
&= \nabla_\theta \mathcal{L}_{GM}(\theta).
\end{aligned}$$

For optimality, Proposition 1 implies that for each $t, h$, the minimizer satisfies:

$$u_t^\theta(x|h) = \mathbb{E}[Y|X = x] = u_t(x|h) \quad \text{a.s. over } p_t(x|h) > 0. \tag{36}$$

$\qquad\square$

## A.2 CONTRASTING CGFM WITH DISCRIMINATIVE MODELS FOR RESIDUAL LEARNING

Another question arises: **Does CGFM truly learn the residual patterns? What if we replace it with a discriminative model—how would the performance differ?** Given an auxiliary model's output and the target value, we can obtain the residual. CGFM naturally constructs a probabilistic path from the auxiliary distribution to the target distribution, thereby learning their transition dynamics. In contrast, a discriminative model (here referring to models that learn deterministic input-to-output

mappings) typically treats the auxiliary output as input and directly learns a mapping to the residual or target value.

To further verify CGFM's capability in learning residual patterns, we compare its performance with that of a discriminative model on the residual correction task. The discriminative model learns a deterministic mapping from the auxiliary model's output to the target (or residual), while CGFM explicitly models the distributional transformation from the auxiliary distribution to the target distribution through a continuous probabilistic path. Here, the discriminative model is defined as RLinear : it takes the RLinear output (auxiliary output) as input, employs a parameterized structure based on RLinear to directly predict the residual $y = x_{\text{target}} - x_{\text{aux}}$, and adds the predicted residual back to the RLinear output to obtain the final prediction. Essentially, it performs deterministic correction using RLinear's parameterized mapping.

Experimental results (Table 5) consistently show that the discriminative model yields higher MSE across all datasets compared to RLinear with CGFM, highlighting inherent limitations of the discriminative paradigm in residual learning—limitations that can be rigorously explained through mathematical analysis of the two models' core designs.

First, the discriminative model's optimization target is inherently constrained to fitting only the first-order moment (conditional mean) of residuals, leading to the loss of critical distributional information.

Formally, let the discriminative model's loss function (based on RLinear's parameterization) be the standard mean squared error (MSE) between the true residual $\epsilon = X_1 - X_0$ and the model's prediction $f_\theta(X_0)$:

$$\mathcal{L}_{\text{disc}}(\theta) = \mathbb{E}_{X_0, X_1} \left[ \|\epsilon - f_\theta(X_0)\|^2 \right], \tag{37}$$

where $X_0$ denotes the auxiliary output (RLinear's prediction), $X_1$ denotes the target value, and $f_\theta(X_0)$ denotes the RLinear-based parameterized function for residual prediction.

We decompose the joint expectation into a nested expectation over $X_0$ and the conditional expectation over $X_1 \mid X_0$:

$$\mathcal{L}_{\text{disc}}(\theta) = \mathbb{E}_{X_0} \left[ \mathbb{E}_{X_1 \mid X_0} \left[ \|\epsilon - f_\theta(X_0)\|^2 \mid X_0 \right] \right]. \tag{38}$$

Expanding the squared term inside the inner expectation:

$$\|\epsilon - f_\theta(X_0)\|^2 = \|\epsilon - \mathbb{E}[\epsilon \mid X_0] + \mathbb{E}[\epsilon \mid X_0] - f_\theta(X_0)\|^2. \tag{39}$$

By the linearity of expectation and the definition of conditional variance, the cross-term vanishes:

$$\mathbb{E}_{X_1 \mid X_0} \left[ (\epsilon - \mathbb{E}[\epsilon \mid X_0])^\top (\mathbb{E}[\epsilon \mid X_0] - f_\theta(X_0)) \mid X_0 \right] = 0. \tag{40}$$

This is because $\mathbb{E}[\epsilon - \mathbb{E}[\epsilon \mid X_0] \mid X_0] = 0$, and $\mathbb{E}[\epsilon \mid X_0] - f_\theta(X_0)$ is a function of $X_0$ (constant with respect to $X_1 \mid X_0$).

then, the loss function decomposes into two terms:

$$\mathcal{L}_{\text{disc}}(\theta) = \mathbb{E}_{X_0} \left[ \|\mathbb{E}[\epsilon \mid X_0] - f_\theta(X_0)\|^2 + \mathbb{V}[\epsilon \mid X_0] \right], \tag{41}$$

where $\mathbb{V}[\epsilon \mid X_0] = \mathbb{E}_{X_1 \mid X_0} \left[ \|\epsilon - \mathbb{E}[\epsilon \mid X_0]\|^2 \mid X_0 \right]$ is the conditional variance of the residual (a non-negative term independent of the model parameter $\theta$).

From this decomposition, we draw two critical conclusions:

1. The Irreducible Error: The term $\mathbb{E}_{X_0} \left[ \mathbb{V}[\epsilon \mid X_0] \right]$ is independent of the model parameters $\theta$. This conditional variance is an intrinsic property of the residual distribution that the model cannot reduce or eliminate.

2. The Target is the Mean: The only term that can be minimized is $\mathbb{E}_{X_0} \left[ \|\mathbb{E}[\epsilon \mid X_0] - f_\theta(X_0)\|^2 \right]$. As a result, the global minimizer of $\mathcal{L}_{\text{disc}}(\theta)$ is proven to be $f_\theta^*(X_0) = \mathbb{E}[\epsilon \mid X_0]$. This means the discriminative model's optimal prediction is fundamentally limited to the conditional mean of the

Table 5: Ablation Study on Mean Squared Error (MSE)

| Model | Weather | Traffic | ETTh1 | ETTh2 | ETTm1 | ETTm2 | Exchange |
|---|---|---|---|---|---|---|---|
| **Rlinear with CGFM** | 0.152 | 0.412 | 0.363 | 0.275 | 0.307 | 0.167 | 0.081 |
| **Rlinear with Discriminative Corrector** | 0.195 | 0.642 | 0.383 | 0.287 | 0.367 | 0.183 | 0.094 |
| **Rlinear (baseline)** | 0.189 | 0.632 | 0.382 | 0.290 | 0.359 | 0.182 | 0.095 |

residual, ignoring any higher-order moments (e.g., variance, skewness) or non-trivial distributional structures.

In stark contrast, Proposition 4.2 of the main text proves that CGFM, under noise smoothing and two-sided coupling conditions, is mathematically equivalent to learning the full probabilistic characteristics of residuals $\epsilon = X_1 - X_0$. The CGFM loss function targets the velocity field of residual evolution:

$$\mathcal{L}_{\text{CGFM}}(\theta) = \mathbb{E}\left[\left\|g_t(X_0, X_1) - u_t^\theta(X_t)\right\|^2\right], \tag{42}$$

where $g_t(X_0, X_1) = \dot\alpha_t X_1 + \dot\beta_t X_0$ encodes the dynamic evolution of the entire residual distribution. This design ties the learning process to the full distribution of $\epsilon$, rather than a single moment—an advantage mathematically unattainable for discriminative models.

Second, the discriminative model fails to incorporate temporal context, leading to incomplete conditional modeling. Its input is restricted to the auxiliary output $X_0$, so its prediction $f_\theta(X_0)$ (and the underlying $\mathbb{E}[\epsilon \mid X_0]$) is independent of historical data $H$. However, residual patterns in time series are highly context-dependent. In contrast, CGFM's two-sided conditional design binds both the source distribution $p(X_0 \mid h)$ (RLinear's output distribution) and target distribution $q(X_1 \mid h)$ to shared historical data, constructing context-aware probability paths:

$$p_{t|H}(x \mid h) = \int p_{t|0,1,H}(x \mid x_0, x_1, h)p(x_0 \mid h)q(x_1 \mid h)dx_0 dx_1. \tag{43}$$

Its velocity field further integrates $H$ to capture temporal dependencies:

$$u_t(x \mid h) = \mathbb{E}\left[\dot\alpha_t X_1 + \dot\beta_t X_0 \mid X_t = x, H = h\right]. \tag{44}$$

This ensures residual corrections align with historical trends, while the discriminative model's context-agnostic $\mathbb{E}[\epsilon \mid X_0]$ averages over dynamic patterns, leading to suboptimal performance.

Third, the discriminative model faces an unbreakable lower bound of expected risk due to its point-estimation nature. And the minimal achievable expected prediction error of the discriminative model is:

$$\min_\theta \mathcal{L}_{\text{disc}}(\theta) = \mathbb{E}_{X_0}\left[\mathbb{V}[\epsilon \mid X_0]\right], \tag{45}$$

which is the average conditional variance of residuals—a non-negative value that is inherent to point estimation (equality holds only if residuals are deterministic). For time series with heteroscedastic residuals, this bound results in unavoidable prediction noise. CGFM circumvents this limitation by optimizing distributional transformations. This explains why CGFM consistently outperforms the discriminative model, even when RLinear provides smooth initial predictions. So the performance gap stems from CGFM's fundamental advantage in residual modeling: unlike the RLinear-based discriminative model, which is mathematically constrained to fitting only the conditional mean of residuals, CGFM leverages flow matching to learn the full probabilistic structure of residuals and their temporal dependencies—validating that explicit residual distribution learning is critical for improving time series forecasting accuracy.

Experimental results are summarized in Table 5. The discriminative model's mean squared error (MSE) is consistently higher than that of "Rlinear with CGFM" across all datasets:

These observations naturally indicate that it is challenging for a discriminative model to predict the gap from the true value when provided with only a single value, as it often makes errors and leads to poorer results. This further underscores the significance of residual learning in CGFM.

---

**Algorithm 2** $X_1$-Prediction Sampling.

---

1: **Input:** History sample $h$, smoothing level $\sigma$, $X_1$-Prediction network $u_t^\theta$, source mode: noise or auxiliary output, prediction objective $g_t(x_0, x_1) = x_1$, time grid $t = [t_0, t_1, \ldots, t_N]$, where $t_0 = 0$ and $t_N = 1$.

2: $h \sim p_H$

3: $x_0 \sim p(x_0|h)$

4: **if** source mode == auxiliary output **then**

5: $\quad \varepsilon \sim \mathcal{N}(0, I)$

6: $\quad x_0 \leftarrow x_0 + \sigma\varepsilon$

7: **end if**

8: **for** $i \leftarrow 0$ to $N - 1$ **do**

9: $\quad t_i \leftarrow t[i], t_{i+1} \leftarrow t[i+1]$

10: $\quad \Delta t_i \leftarrow t_{i+1} - t_i$

11: $\quad u_{t_i} \leftarrow \frac{\dot{\beta}_{t_i}}{\beta_{t_i}} \cdot x_{t_i} + \left( \dot{\alpha}_{t_i} - \frac{\alpha_{t_i} \dot{\beta}_{t_i}}{\beta_{t_i}} \right) \cdot u_t^\theta(x_{t_i}|h)$

12: $\quad x_{\mathrm{mid}} \leftarrow x_{t_i} + \frac{\Delta t_i}{2} \cdot u_{t_i}$

13: $\quad u_{\mathrm{mid}} \leftarrow \frac{\dot{\beta}_{t_i + \Delta t_i/2}}{\beta_{t_i + \Delta t_i/2}} \cdot x_{\mathrm{mid}} + \left( \dot{\alpha}_{t_i + \Delta t_i/2} - \frac{\alpha_{t_i + \Delta t_i/2} \dot{\beta}_{t_i + \Delta t_i/2}}{\beta_{t_i + \Delta t_i/2}} \right) \cdot u_t^\theta(x_{\mathrm{mid}}|h)$

14: $\quad x_{t_{i+1}} \leftarrow x_{t_i} + \Delta t_i \cdot u_{\mathrm{mid}}$

15: **end for**

16: **Output:** $x_1$

---

## A.3 TRAINING AND SAMPLING ALGORITHM

---

**Algorithm 1** CGFM Training

---

1: **Input:** History distribution $p_H$, path parameters $\alpha_t, \beta_t$, smoothing level $\sigma$, network $u_t^\theta$, source distribution $p(x_0 \mid h)$, source mode: noise or auxiliary output, prediction objective $g_t$, and target distribution $q(x_1 \mid h)$.

2: **repeat**

3: $\quad h \sim p_H$

4: $\quad x_0 \sim p(x_0|h); \; x_1 \sim q(x_1|h)$

5: $\quad$ **if** source mode == auxiliary output **then**

6: $\quad\quad \varepsilon \sim \mathcal{N}(0, I)$

7: $\quad\quad x_0 \leftarrow x_0 + \sigma\varepsilon$

8: $\quad$ **end if**

9: $\quad t \sim \mathcal{U}(0, 1)$

10: $\quad x_t \leftarrow \alpha_t x_1 + \beta_t x_0$

11: $\quad \mathcal{L}_{CGM}(\theta) \leftarrow \left\| g_t(x_0, x_1) - u_t^\theta(x_t|h) \right\|^2$

12: $\quad \theta \leftarrow \mathrm{Update}(\theta, \nabla_\theta \mathcal{L}_{CGM}(\theta))$

13: **until** converged

14: **Return:** $u_t^\theta$

---

The complete methodology, detailing the training and sampling procedures for CGFM, is presented in Algorithm 1 (Training) and Algorithm 2 (Sampling).

## A.4 MORE RELATED WORK

### A.4.1 DEEP LEARNING FOR TIME SERIES FORECASTING: FROM EARLY PARADIGMS TO FLOW MATCHING

Deep learning has propelled remarkable progress in time series forecasting, with a diverse array of paradigms evolving to tackle the challenges of modeling temporal dependencies. RNN-based models (Salinas et al., 2020; Du et al., 2021), characterized by their recurrent architectures, excel at capturing sequential patterns but are plagued by error propagation when dealing with long-horizon forecasting tasks. CNN-based models (Wang et al., 2023; Franceschi et al., 2019; Liu et al., 2022a)

leverage convolutional kernels to extract local features; however, their receptive fields are inherently constrained, restricting their capacity to model long-range dependencies effectively. Transformer-based models (Zhou et al., 2021; Liu et al., 2023; Chen et al., 2021; Lu et al., 2025; Shi et al., 2025; Liu et al., 2025; Chen et al., 2025; Wang et al., 2025b), empowered by self-attention mechanisms, have attained state-of-the-art performance by capturing global temporal correlations, though their quadratic time and memory complexity presents significant scalability hurdles. In contrast, MLP-based models (Zeng et al.; Yi et al., 2024; Wang et al., 2024) , with their simpler architectural designs, have recently demonstrated promising results in modeling cross-domain relationships across both time and frequency domains, offering a lightweight alternative to more complex transformer-based approaches. Similarly, KAN-based models (Huang et al., 2025), spiking neural networks (SHIBO et al., 2025), as well as other works related to dynamical systems (Zheng et al., 2025; Wang et al., 2025a), have also emerged.

As generative modeling gained prominence, diffusion-based models emerged as powerful tools in time series forecasting. These models (Shen & Kwok, 2023a; Yuan & Qiao, 2024; Alcaraz & Strodthoff, 2022; Zhong et al., 2025; Barancikova et al., 2025; Li et al., 2025) utilize iterative denoising processes to transform noise into realistic future sequences, excelling in capturing complex predictive distributions and enabling conditional generation . Their key strength lies in modeling uncertainty and generating diverse forecasts, making them well-suited for scenarios demanding probabilistic outputs. Nevertheless, diffusion models often suffer from slow sampling due to their stepwise denoising process. Moreover, their reliance on simple Gaussian priors (Rasul et al., 2021) and rigid constraints on sampling trajectories (Ho et al., 2020) limit their ability to capture the intricate temporal structures inherent in real-world time series data .

Flow matching (Lipman et al., 2023), a state-of-the-art generative framework, addresses these limitations and offers distinct advantages for time series forecasting. Compared to diffusion models, flow matching (Albergo & Vanden-Eijnden, 2023) simplifies the generative process by learning continuous velocity fields that directly map source to target distributions, enabling more efficient sampling without the need for iterative denoising . It provides greater flexibility in the choice of initial distributions—such as leveraging auxiliary model outputs instead of simple Gaussian noise—and in the design of sampling trajectories, which is critical for aligning with the temporal characteristics of time series data (Liu et al., 2022b).

### A.4.2 RESIDUAL MODELING

Residuals have long been studied in time series forecasting, but their treatment varies significantly across methods. Traditional statistical models (e.g., ARIMA (Shumway & Stoffer, 2017)) use residuals to diagnose model fit but rarely model their probabilistic structure, treating them as white noise. Deep learning models, such as residual networks (ResNets (He et al., 2016)) and Transformer variants, use residual connections to mitigate training issues (e.g., vanishing gradients) but still reduce residuals to optimization targets.

CGFM by explicitly learning the probabilistic characteristics of residuals via flow matching, treating them as a bridge between auxiliary model predictions and true future values. This allows CGFM to address systematic biases and non-trivial residual structures, which are common in real-world time series but overlooked by existing methods.

### A.4.3 FLOW MATCHING

Flow matching (Lipman et al., 2023) has developed through several key advancements to enhance generative modeling capabilities. Rectified Flow (Esser et al., 2024) accelerates the generative process by learning "straight paths" in data space, simplifying trajectory optimization. However, it relies on specialized training schemes to prevent path crossing, which limits its adaptability to time series data with dynamic temporal structures . Stochastic Interpolants (Albergo et al., 2023) introduce affine conditional flows to expand the space of feasible probability paths, offering greater flexibility in mapping between source and target distributions. However, they differ from flow matching in terms of modeling, and their loss functions are also different. Although they use affine paths, directly migrating these paths to flow matching will encounter theoretical issues. OT-CFM (Tong et al., 2024) leverages optimal transport plans to refine distribution alignment, reducing the gap between source and target. However, the complex computational overhead of optimal transport and rigid transport

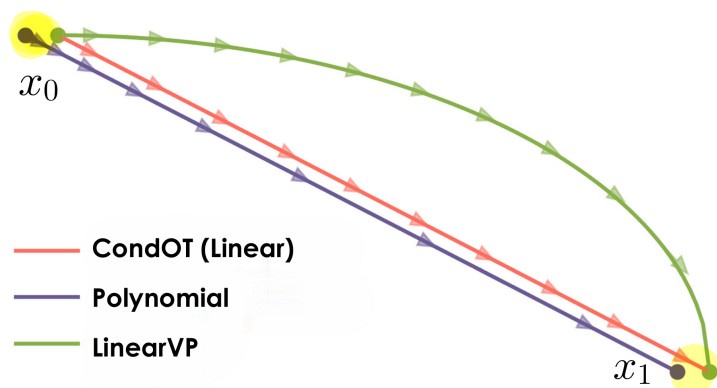

Figure 7: The condOT path is linear with constant speed, the polynomial path is linear with increasing speed, and the LinearVP path is a curve with increasing speed.

plans hinder its scalability for high-dimensional time series, limiting adaptability to dynamic temporal patterns .

Our CGFM naturally avoids path crossing through two-sided conditional paths (where both the source and target are conditioned on shared historical data) and affine path designs, thus eliminating the need for specialized training schemes . By utilizing the auxiliary model's predictions as the source distribution—which is closer to the target distribution than generic priors—it effectively reduces the difficulty of learning. Moreover, CGFM incorporates historical data into both the construction of probability paths and the learning of velocity fields, ensuring temporal consistency and preserving critical temporal dependencies.

## A.5 THE PARAMETERIZATIONS OF AFFINE PATH $\alpha_t$ AND $\beta_t$

The parameterizations of $\alpha_t$ and $\beta_t$ for the schedulers used in this work are defined as follows:

$$
\begin{aligned}
\text{CondOT:} \quad & \alpha_t = t, \quad \beta_t = 1 - t, \\
\text{Poly-n:} \quad & \alpha_t = t^n, \quad \beta_t = 1 - t^n, \\
\text{LinearVP:} \quad & \alpha_t = t, \quad \beta_t = \sqrt{1 - t^2}, \\
\text{Cosine:} \quad & \alpha_t = \sin\left(\frac{\pi t}{2}\right), \quad \beta_t = \cos\left(\frac{\pi t}{2}\right).
\end{aligned}
\tag{46}
$$

Here, $t \in [0, 1]$ represents the normalized time step, and $n$ in the Poly-n scheduler is a positive hyperparameter controlling the polynomial degree. Each parameterization is designed to encode distinct behaviors in the dynamics of the scheduling process.

Figure 7 illustrates the velocity vectors corresponding to different paths. CondOT exhibits uniformly distributed arrows, while Poly-3 and LinearVP display sparse arrows in the early stages and denser ones later. This dynamic velocity pattern ensures that the model avoids "large-step" updates near $X_0$, thereby reducing the risk of local optima and improving convergence stability.

## A.6 EVALUATION METRICS FOR TIME SERIES FORECASTING

### A.6.1 MEAN SQUARED ERROR (MSE)

MSE quantifies the average of squared differences between predicted values and ground-truths, emphasizing the impact of larger errors. It is defined as:

$$
\text{MSE} = \frac{1}{C \times L} \sum_{c=1}^{C} \sum_{t=1}^{L} (r_t^c - \hat{r}_t^c)^2
\tag{47}
$$

Table 6: Ablation Study on Mean Squared Error (MSE)

| Model | Weather | Traffic | ETTh1 | ETTh2 | ETTm1 | ETTm2 | Exchange |
|---|---|---|---|---|---|---|---|
| **CGFM** | 0.152 | 0.412 | 0.363 | 0.275 | 0.307 | 0.167 | 0.081 |
| **Without Flow Matching** | 0.193 | 0.635 | 0.386 | 0.299 | 0.368 | 0.192 | 0.096 |
| **CGFM Independent Coupling** | 0.158 | 0.424 | 0.370 | 0.283 | 0.314 | 0.171 | 0.084 |
| **CGFM One-Sided Condition** | 0.161 | 0.430 | 0.373 | 0.286 | 0.317 | 0.173 | 0.085 |

where: $C$ = number of variates (e.g., multiple time series variables), $L$ = length of the time series, $r_t^c$ = ground-truth value of the $c$-th variate at timestep $t$, $\hat{r}_t^c$ = predicted value of the $c$-th variate at timestep $t$.

Smaller MSE indicates closer alignment between predictions and ground-truths. The squaring operation amplifies the penalty for large errors, making MSE sensitive to outliers.

### A.6.2 MEAN ABSOLUTE ERROR (MAE)

MAE measures the average magnitude of absolute differences between predictions and observations, avoiding the outlier bias of MSE. Its formulation is:

$$\text{MAE} = \frac{1}{C \times L} \sum_{c=1}^{C} \sum_{t=1}^{L} |r_t^c - \hat{r}_t^c| \tag{48}$$

MAE provides a straightforward interpretation of "average error size" and is robust to extreme outliers, though less sensitive to their impact compared to MSE.

### A.6.3 CONTINUOUS RANKED PROBABILITY SCORE (CRPS)

CRPS evaluates the quality of probabilistic forecasts by comparing the predicted cumulative distribution function (CDF) $F$ with the empirical CDF of observations $r$. It is defined as:

$$\text{CRPS} = \int_{\mathbb{R}} \left( F(z) - \mathbb{I}_{\{r \leq z\}} \right)^2 dz \tag{49}$$

where $\mathbb{I}_{\{r \leq z\}}$ is an indicator function (1 if $r \leq z$, 0 otherwise). For practical computation, use the empirical CDF from simulated samples:

$$\hat{F}(z) = \frac{1}{n} \sum_{i=1}^{n} \mathbb{I}_{\{R_i \leq z\}} \tag{50}$$

($R_1, \ldots, R_n$ are simulated samples of the predicted distribution). CRPS rewards forecasts where $F$ matches the true data distribution (minimum value = 0). Lower CRPS indicates better-calibrated uncertainty.

### A.7 ABLATION STUDY

### A.7.1 ABLATION ON CGFM COMPONENTS

The proposed CGFM model consists of four main components: the choice of the initial distribution via the Auxiliary Model, the selection of the probability path, the design of the prediction functions, and the application of the flow matching method. The ablation study on the choice of the Auxiliary Model is presented in Table 2 of the main text. The ablation study on the probability path is shown in Table 3 of the main text, while the impact of the prediction functions is detailed in Table 4 of the main text. Since both the probability path and prediction functions are integral to the flow matching framework, they cannot be entirely removed from the model.

Thus, as shown in Table 6, we conducted experiments including the complete CGFM with RLinear, the standalone velocity network (denoted as "Without Flow Matching"), CGFM with independent coupling, and CGFM with one-sided condition (which is equivalent to using a Gaussian initial distribution). The results in the table indicate that each component of CGFM forms an indispensable organic whole, collectively contributing to the improvement of model performance.

### A.7.2 Smooth Parameter for Auxiliary Model

For the ETTh1 dataset, each sample's prediction is treated as a high-dimensional vector ($d = 96 \times 7$), capturing joint information across all channels over the 96-step prediction horizon. To analyze prediction smoothness, Principal Component Analysis (PCA) was applied to the high-dimensional data matrix $\mathbf{X} \in \mathbb{R}^{n \times d}$, where $n$ is the number of samples. The first two principal components (PC1 and PC2) were retained to project the data into a 2D subspace for structural visualization. Temporal continuity was assessed by visualizing the sequential trajectory of samples in the PCA space, with color gradients representing sample indices and lines connecting points. As shown on the left of Figure 4, the PCA trajectory of RLinear predictions closely aligns with the ground truth in both shape and continuity, indicating that RLinear effectively captures the temporal evolution of the underlying physical process. In contrast, iTransformer predictions and TimeDiff predictions exhibit irregular spatial distributions, with discontinuities in inter-point connections. This suggests non-smooth fluctuations in the high-dimensional representation of their predictions.

Table 7: Impact of Smooth Parameter (Noise Scale) on Model Performance (MSE) on ETTh1 Dataset.

| Noise Scale | 0.1 | 1 | 5 | 10 | 20 | 30 |
|---|---|---|---|---|---|---|
| **RLinear (MSE)** | 0.364 | 0.363 | 0.369 | 0.367 | 0.374 | 0.371 |
| **iTransformer (MSE)** | 0.370 | 0.368 | 0.379 | 0.460 | 0.466 | 0.431 |

### A.8 Robust Result

We conducted a statistically robust variance analysis across ten different random seeds under the conditions of Table 2 in the main text (Performance without Auxiliary Model). Specifically, the Mean Squared Error (MSE) ± Standard Deviation (SD) values for each task were evaluated on seven benchmark datasets. Lower MSE values indicate better performance, as shown in Table 8. The results fully demonstrate the effectiveness of flow matching.

### A.9 Comparison of Coupling Methods

As illustrated in Figure 8, a critical distinction exists between independent and conditional independent coupling in our proposed framework, with profound implications for the non-crossing property of the probability flow. The left panel shows that independent coupling, which constructs a path by treating the source distribution $p(x_0)$ and the target distribution $q(x_1)$ as unrelated, may produce crossing paths. This is a consequence of the underlying mathematical problem: without a guiding context, the flow may lack the necessary smoothness to ensure a unique solution to the probability flow ODE. In such a scenario, the velocity field may not satisfy the local Lipschitz continuity condition, a fundamental requirement for the uniqueness of ODE solutions. As a result, distinct initial conditions $X_0^{(1)}$ and $X_0^{(2)}$ could converge to the same point $z$ at an intermediate time $t$, leading to ambiguous mappings and information loss, which is particularly detrimental in time series forecasting.

In contrast, our CGFM method employs conditional independent coupling, as shown in the right panel, which robustly addresses this issue. By conditioning both $p(x_0|h)$ and $q(x_1|h)$ on the shared historical data $h$, we introduce a unified context that inherently structures the mapping from the auxiliary model's predictions to the true future values. This design ensures that the derived velocity field is sufficiently smooth, which in turn guarantees the non-crossing property of the probability paths (as rigorously established in Proposition 4.5). This approach is not merely a theoretical exercise; it is a practical solution that ensures the model's velocity field satisfies the key mathematical properties required for a well-behaved and stable flow. By providing a coherent, history-driven context, CGFM transforms a potentially chaotic and ill-posed transformation into an orderly and predictable flow, which is essential for learning a consistent and effective mapping in complex time series forecasting tasks.

### A.10 CRPS Comparison

Given that CRPS is a metric tailored for probabilistic forecasting, it is not applicable to point forecasting models like Rlinear and iTransformer in the absence of prior assumptions. Consequently,

Table 8: Robust Result

| | Weather | Traffic | ETTh1 | ETTh2 | ETTm1 | ETTm2 | Exchange |
|---|---|---|---|---|---|---|---|
| **CGFM** | $0.161 \pm 0.0002$ | $0.430 \pm 0.0005$ | $0.373 \pm 0.0003$ | $0.286 \pm 0.0002$ | $0.317 \pm 0.0001$ | $0.173 \pm 0.0005$ | $0.085 \pm 0.0001$ |

Figure 8: Difference between independent coupling and conditional independent coupling. In each figure, the left end represents $X_0$ and the right end represents $X_1$. It can be observed that conditional independent coupling helps reduce path crossings and establishes a one-to-one correspondence between $X_0$ and $X_1$.

our comparison is restricted to the following probabilistic forecasting models: TimeDiff (Shen & Kwok, 2023b), CSDI (Tashiro et al., 2021), SSSD (Alcaraz & Strodthoff, 2023), $D^3$VAE (Li et al., 2023a), TimeGrad (Rasul et al., 2021), and TSDiff (Kollovieh et al., 2023).

Table 9: Comparison of CRPS ($\downarrow$) on Diverse Time Series Datasets. CGFM consistently improves the performance of various baseline models (lower is better).

| Model | | Weather | Traffic | ETTh1 | ETTh2 | ETTm1 | ETTm2 | Exchange |
|---|---|---|---|---|---|---|---|---|
| **TimeDiff** | without CGFM | 0.072 | 0.238 | 0.392 | 0.237 | 0.324 | 0.196 | 0.038 |
| | **+CGFM** | **0.063** | **0.212** | **0.334** | **0.181** | **0.279** | **0.139** | **0.026** |
| **CSDI** | without CGFM | 0.095 | — | 0.567 | 0.314 | 0.341 | 0.211 | 0.045 |
| | **+CGFM** | **0.078** | — | **0.432** | **0.267** | **0.298** | **0.176** | **0.034** |
| **SSSD** | without CGFM | 0.081 | 0.251 | 0.416 | 0.286 | 0.344 | 0.224 | 0.041 |
| | **+CGFM** | **0.075** | **0.234** | **0.378** | **0.249** | **0.302** | **0.189** | **0.031** |
| $D^3$**VAE** | without CGFM | 0.102 | 0.242 | 0.546 | 0.305 | 0.395 | 0.246 | 0.052 |
| | **+CGFM** | **0.084** | **0.221** | **0.429** | **0.271** | **0.352** | **0.213** | **0.039** |
| **TimeGrad** | without CGFM | 0.165 | 0.256 | 0.589 | 0.451 | 0.649 | 0.528 | 0.077 |
| | **+CGFM** | **0.092** | **0.239** | **0.457** | **0.322** | **0.408** | **0.276** | **0.046** |
| **TSDiff** | without CGFM | 0.123 | 0.351 | 0.514 | 0.420 | 0.434 | 0.445 | 0.091 |
| | **+CGFM** | **0.082** | **0.264** | **0.367** | **0.298** | **0.391** | **0.298** | **0.051** |

From Table 9, it can be observed that CGFM consistently boosts the probabilistic forecasting performance of all baseline models, with lower CRPS values achieved across nearly all datasets. For every evaluated model—including TimeDiff, CSDI, SSSD, $D^3$VAE, TimeGrad, and TSDiff—the variant with **+CGFM** outperforms its "without CGFM" counterpart. The improvement is robust: even for models with higher baseline CRPS (e.g., TimeGrad and TSDiff), CGFM drives substantial reductions. This confirms CGFM's strong generality and effectiveness in enhancing diverse probabilistic time series forecasting models.

### A.11 The Relationship Between Flow Matching and Diffusion Models

#### A.11.1 Time setup

The first distinction lies in the time setup between diffusion models and flow matching. In flow matching, time ranges from 0 to 1, where the source distribution (possibly noise) is typically at $t = 0$, and the target distribution (the desired outcome) is at $t = 1$. Conversely, for diffusion models, the timeframe extends from $+\infty$ to 0. At $t = +\infty$, the diffusion model represents noise, whereas at $t = 0$, it represents the target distribution.

Thus, a strictly decreasing function $k$ can be employed here to map $(0, 1]$ to $[0, +\infty)$ such that $k(1) = 0$ and $k(t)$ approaches $+\infty$ as $t$ approaches 0.

#### A.11.2 Noise Addition Process

Denoising diffusion models fundamentally operate on the concept of crafting a forward process that deteriorates the data distribution. This notion aligns with a specific construction of a probability path, as employed in Flow Matching (FM). The forward process, denoted as $X_r$, is articulated through the Stochastic Differential Equation (SDE)

$$dX_r = a_r(X_r)dr + g_r dW_r, X_0 \sim q, \tag{51}$$

where $q$ embodies the data distribution, $W_r$ is a Brownian motion, $a : R \times R^d \to R^d$ is a velocity field (referred to as drift in SDE contexts), and $g : R \to R_{\geq 0}$ is a diffusion coefficient.

Each SDE introduces a conditional probability path and a marginal probability path as follows:

$$\tilde{p}_{r|0}(x|z) = P[X_t = x|X_0 = z], \quad \tilde{p}_r(x) = P[X_t = x] \tag{52}$$

$$p_{t|1}(x|z) = \tilde{p}_{k(t)|0}(x|z), \quad p_t(x) = \tilde{p}_{k(t)}(x) \tag{53}$$

In Equation 53, time is reparameterized into the FM time parameterization. Evidently, $p_{t|1}(x|z)$ provides a conditional probability path. Moreover, the forward process is structured such that for sufficiently large $R$, the distribution of $X_R$ approximates a Gaussian. The conditional probability path $p_{t|1}(x|z)$ represents the distribution of the forward process SDE when initialized with $X_0 = z$.

#### A.11.3 Training and Sampling

The loss function for Denoising Score Matching is fundamental for training diffusion models and is expressed as follows:

$$
\begin{aligned}
L_{\text{CM}}(\theta) &= \mathbb{E}_{t,Z\sim q,X_0\sim p} \left\| x_{0|t}^\theta(\alpha_t X_0 + \sigma_t Z) - X_0 \right\|^2 \\
&= \mathbb{E}_{t,Z\sim q,X_t\sim p_{t|1}(\cdot|Z)} \sigma_t^2 \left\| s_t^\theta(X_t) - \left[ -\frac{1}{\sigma_t^2}(X_t - \alpha_t Z) \right] \right\|^2 \\
&= \mathbb{E}_{t,Z\sim q,X_t\sim p_{t|1}(\cdot|Z)} \sigma_t^2 \left\| s_t^\theta(X_t) - \nabla \log p_{t|1}(X_t|Z) \right\|^2 .
\end{aligned}
\tag{54}
$$

The first transformation involves reparameterizing the neural network as $s_t^\theta = -x_{0|t}^\theta/\sigma_t$. The optimal parameter $\theta^*$ satisfies $s_t^{\theta^*}(x) = -\frac{1}{\sigma_t}\mathbb{E}[X_0 \mid X_t = x] = \nabla \log p_t(x)$.

In the context of sampling, we explore the relationship to sampling from FM or GM models. For deterministic sampling, if the diffusion model is considered as an FM model, sampling is performed by drawing from the marginal vector field, expressed via the score function for Gaussian paths:

$$u_t(x) = \frac{\dot{\alpha}_t}{\alpha_t}x - \frac{\dot{\sigma}_t}{\sigma_t} - \frac{\sigma_t^2}{2}\frac{\dot{\alpha}_t}{\alpha_t}\nabla \log p_t(x). \tag{55}$$

We derive the equivalent identity:

$$u_t(x) = \dot{k}(t)\alpha_t x - \frac{g_t^2}{2}\nabla \log p_t(x). \tag{56}$$

This can be directly inserted into the Continuity Equation. The corresponding ODE, also known as the Probability Flow ODE, is given by:

$$dX_t = \dot{k}(t)\alpha_t X_t - \frac{g_t^2}{2}s_t^\theta(X_t)dt, \tag{57}$$

where $s_t^\theta(x) = \nabla \log p_t(x)$ is the learned score function. The notation used here for ODEs is common in SDEs, which becomes clear later. The addition of the term $\dot{k}(t)$ is due to time reparameterization.

For stochastic sampling with SDEs, adding Langevin dynamics to any CTMP generative model results in a model following the same probability path. Applying this to the Probability Flow ODE yields a family of SDEs generating the probability path $p_t$:

$$dX_t = \dot{k}(t)\alpha_t X_t + \left(\beta_t^2 - \frac{\dot{k}(t)g_t^2}{2}\right)\nabla \log p_t(X_t)dt + \beta_t dW_t. \tag{58}$$

This results in stochastic sampling of a diffusion model. Theoretically, all models yield the same marginals for each $\beta_t \geq 0$. Practically, simulating the SDE:

$$dX_t = \dot{k}(t)\alpha_t X_t + \left(\beta_t^2 - \frac{\dot{k}(t)g_t^2}{2}\right)s_t^\theta(X_t)dt + \beta_t dW_t \tag{59}$$

Using a trained network $s_t^\theta$ involves estimation errors (due to imperfect training of $s_t^\theta$) and simulation errors (due to imperfect sampling of the underlying SDE). Consequently, optimal noise levels $\beta_t$ must be determined. ODE sampling for a Gaussian source with independent coupling, specified $\alpha_t$, $\sigma_t$, and score parameterization equates sampling from a diffusion model using the Probability Flow ODE to sampling from a Flow Matching (FM) model.

### A.12 PREDICTION RESULTS VISUALIZATION

The visualization of the results for rlinear and itransformer, both with and without CGFM.

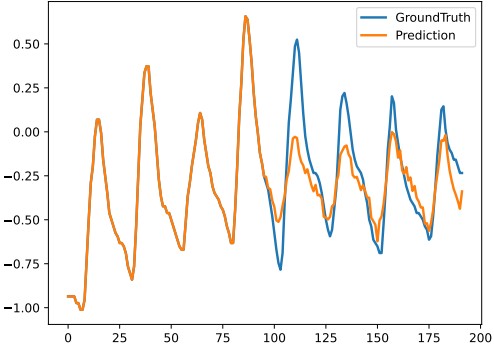 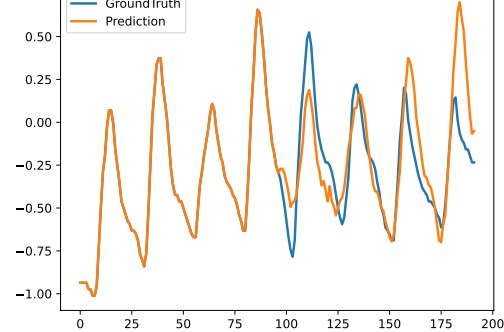

Figure 9: Rlinear Without CGFM          Figure 10: Rlinear With CGFM

Figure 11: Comparison of Prediction Results on ETTh2

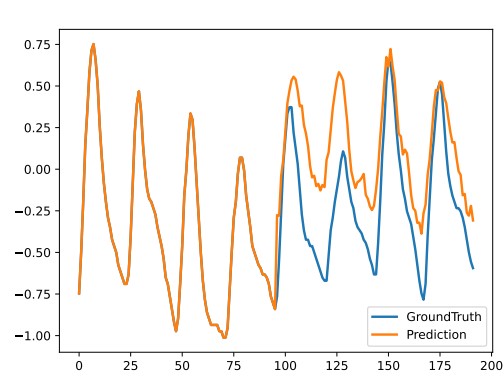

Figure 12: iTransformer Without CGFM

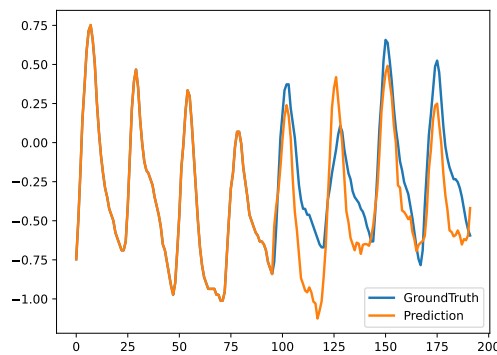

Figure 13: iTransformer With CGFM

Figure 14: Comparison of Prediction Results on ETTh2

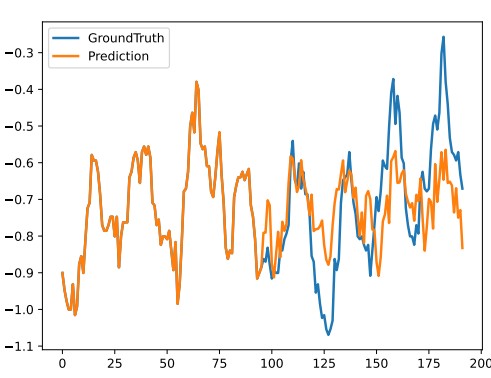

Figure 15: Rlinear Without CGFM

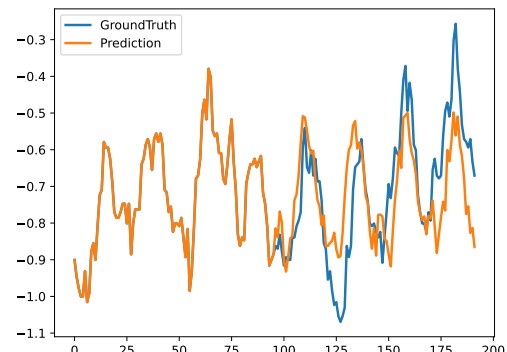

Figure 16: Rlinear With CGFM

Figure 17: Comparison of Prediction Results on ETTh1

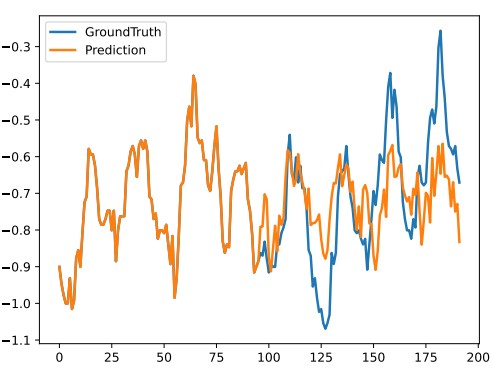

Figure 18: iTransformer Without CGFM

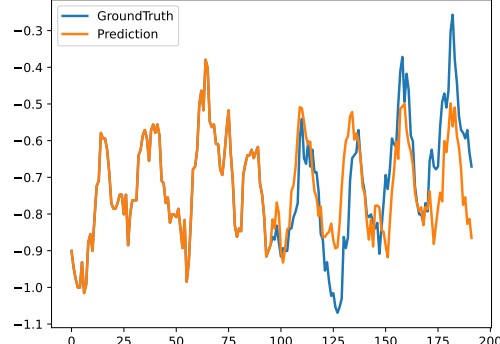

Figure 19: iTransformer With CGFM

Figure 20: Comparison of Prediction Results on ETTh1

## A.13 DATASETS

(1) ETT dataset, which includes ETTh1, ETTh2, ETTm1, and ETTm2, contains data collected from electricity transformers, including load and oil temperature measurements, from July 2016 to July 2018. (2) Electricity dataset contains the hourly electricity consumption of 321 customers from 2012 to 2014. (3) Exchange dataset records the daily exchange rates of eight different countries from 1990 to 2016. (4) Traffic dataset comprises hourly data from the California Department of Transportation, describing the road occupancy rates measured by various sensors on San Francisco Bay Area freeways. (5) Weather dataset, recorded every 10 minutes throughout 2020, contains 21 meteorological indicators, such as air temperature and humidity. We followed standard protocols and split all datasets into training, validation, and test sets in chronological order, with a ratio of 6:2:2 for the ETT dataset and 7:1:2 for the other datasets.

Table 10: Summary of dataset statistics, including dimension, total observations, and sampling frequency.

| Dataset | Dim | # Observations | Freq. |
|---|---|---|---|
| Weather | 21 | 52,696 | 10 mins |
| Traffic | 862 | 17,544 | 1 hour |
| Electricity | 321 | 26,304 | 1 hour |
| ETTh1 | 7 | 17,420 | 1 hour |
| ETTh2 | 7 | 17,420 | 1 hour |
| ETTm1 | 7 | 69,680 | 15 mins |
| ETTm2 | 7 | 69,680 | 15 mins |
| Exchange | 8 | 7,588 | 1 day |

## A.14 INFERENCE EFFICIENCY ANALYSIS

This section compares the inference efficiency of various time series models, focusing on the key metric **Infer per iter** (inference time per iteration, batch=32), where lower values indicate faster inference speed.

Table 11: Inference Efficiency of Various Time Series Models

| Model | Infer per iter |
|---|---|
| CGFM | 0.026s |
| TimeDiff | 0.083s |
| TSDiff | 0.236s |
| TimesNet | 0.035s |
| $D^3$VAE | 0.188s |
| CSDI | 0.521s |
| RLinear | 0.004s |
| iTransformer | 0.009s |
| PatchTST | 0.010s |
| Pathformer | 0.023s |
| Informer | 0.105s |
| Fedformer | 0.216s |
| SSSD | 0.227s |
| Autoformer | 0.295s |
| TimeGrad | 0.978s |

As shown in the results Table 11, CGFM achieves the fastest inference speed among all diffusion-based models, which can be attributed to its simple yet efficient design and the high inference efficiency of flow matching. CGFM also outperforms most Transformer-based models in inference speed,

though it is slightly slower than some lightweight architectures such as RLinear or PatchTST. Overall, CGFM demonstrates relatively fast inference, striking a favorable balance between performance and efficiency.

### A.14.1 CASE STUDY OF RESIDUAL LEARNING

Figures 5 ,6,21,22 are generated based on the 96-to-96 prediction task on the ETTh1 dataset. It compares the performance of Rlinear and iTransformer, both with and without CGFM. Specifically, the prediction results (`preds`) of Rlinear, iTransformer and the corresponding ground truth values (`trues`) are collected first, both of which have dimensions $[N, 96, C]$, where $N$ is the total number of samples and $C = 7$ is the number of channels in ETTh1. For each channel, the mean value across all samples at each time step is calculated: the mean of `preds[:, :, ch]` along the sample dimension (`axis=0`) yields the predicted mean sequence ; the same is done for `trues[:, :, ch]` to obtain the ground truth mean sequence. This approach eliminates individual sample fluctuations and highlights the overall trend.

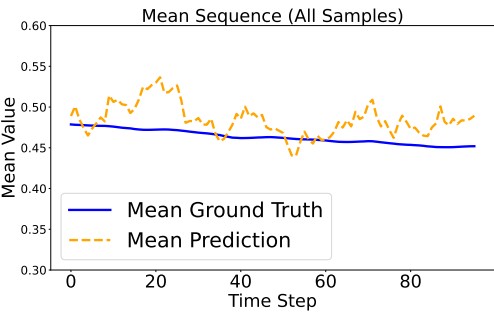 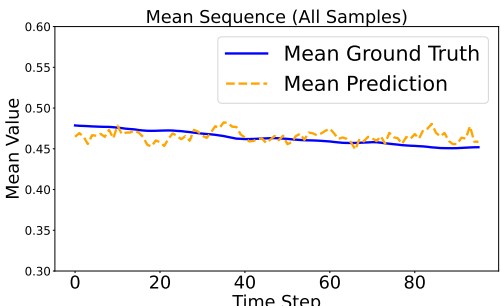

Figure 21: iTransformer Without CGFM      Figure 22: iTransformer With CGFM

Figure 23: Comparison of Mean Sequences of iTransformer (Without vs. With CGFM)

### A.15 $u_t^\theta$ ARCHITECTURE

The proposed architecture integrates temporal embeddings with convolutional operations to effectively model and forecast time series data. Given an input sequence $\mathbf{x} \in \mathbb{R}^{B \times C \times N}$, where $B$ is the batch size, $C$ is the number of input channels, and $N$ is the sequence length, the network processes the data in a sequential manner to produce accurate predictions.

First, the input sequence $\mathbf{x}$ is transformed using an input convolutional layer to extract local temporal features. This operation produces a feature map $\mathbf{H}_{\text{inp}} \in \mathbb{R}^{B \times K \times N}$, where $K$ is the number of output channels:

$$\mathbf{H}_{\text{inp}} = \text{InputConv}(\mathbf{x}).$$

To incorporate temporal information, temporal embeddings $\mathbf{D}_{\text{expand}} \in \mathbb{R}^{B \times D \times N}$ are concatenated with the extracted feature map $\mathbf{H}_{\text{inp}}$, resulting in a fused representation $\mathbf{H}_{\text{cat}} \in \mathbb{R}^{B \times (K+D) \times N}$:

$$\mathbf{H}_{\text{cat}} = \text{Concat}(\mathbf{H}_{\text{inp}}, \mathbf{D}_{\text{expand}}).$$

The concatenated features are passed through an encoder convolutional layer, which transforms them into intermediate representations $\mathbf{H}_{\text{enc}} \in \mathbb{R}^{B \times F \times N}$, where $F$ is the number of output channels of the encoder:

$$\mathbf{H}_{\text{enc}} = \text{EncoderConv}(\mathbf{H}_{\text{cat}}).$$

To incorporate conditional information, external conditional inputs $\mathbf{c} \in \mathbb{R}^{B \times M \times N}$ are projected into a feature space that aligns with the prediction task. This projection is performed using a linear transformation, producing $\mathbf{O} \in \mathbb{R}^{B \times P \times N}$, where $P$ is the projected dimension:

$$\mathbf{O} = \text{Linear}(\mathbf{c}).$$

The encoded features $\mathbf{H}_{\text{enc}}$ and the projected conditional features $\mathbf{O}$ are then concatenated and processed through a combination convolutional layer to produce $\mathbf{H}_{\text{comb}} \in \mathbb{R}^{B \times C' \times N}$, where $C'$ is the number of output channels of the combination layer:

$$\mathbf{H}_{\text{comb}} = \text{CombineConv}(\text{Concat}(\mathbf{H}_{\text{enc}}, \mathbf{O})).$$

Finally, the combined features $\mathbf{H}_{\text{comb}}$ are mapped to the target output space $\mathbf{y} \in \mathbb{R}^{B \times L \times N}$, where $L$ is the number of output variables, using an output convolutional layer:

$$\mathbf{y} = \text{OutputConv}(\mathbf{H}_{\text{comb}}).$$

Throughout the architecture, the SiLU (Sigmoid Linear Unit) activation function is applied to introduce nonlinearity:

$$\text{SiLU}(x) = x \cdot \sigma(x), \quad \sigma(x) = \frac{1}{1 + e^{-x}},$$

where $\sigma(x)$ is the sigmoid function. The use of SiLU improves gradient flow and enhances the network's ability to learn complex temporal patterns.

The design of the Velocity Net architecture ensures a comprehensive integration of temporal embeddings, conditional inputs, and convolution-based feature extraction, making it well-suited for time series forecasting tasks.

### A.16  THE USE OF LARGE LANGUAGE MODELS (LLMS)

In this work, large language models (LLMs) were used exclusively as a general-purpose writing aid. Their role was limited to polishing the paper's language and phrasing—including enhancing the fluency of academic expressions, standardizing technical descriptions (e.g., refining the wording of model formulations and experimental procedures), and sharpening logical coherence between sections.

Notably, LLMs played no role in any core research activities: they did not contribute to research ideation, the design of the Conditional Guided Flow Matching (CGFM) framework, the derivation of theoretical propositions or proofs, experimental design, data analysis, or the drawing of conclusions. As such, LLMs do not qualify as contributors to this research.

We affirm full responsibility for all content of the paper, including text refined by LLMs. We have thoroughly reviewed and verified every section to ensure it is free of plagiarism, factual fabrication, or other forms of scientific misconduct. In compliance with ICLR guidelines, LLMs are not listed as authors. You may include other additional sections here.

