# OpenReview forum: "Conditional Guided Flow Matching: Modeling Prediction Residuals for Enhanced Time Series Forecasting"
_ICLR.cc/2026/Conference — Submitted to ICLR 2026_

### Official Review · Reviewer_knHF · 2025-10-28

**Soundness:** 4
**Presentation:** 3
**Contribution:** 3
**Rating:** 8
**Confidence:** 4

**Summary:**

This paper proposes Conditional Guided Flow Matching, CGFM, a Flow Matching-based framework that enhances time series forecasting by explicitly modeling the full probabilistic structure of prediction residuals. It addresses the limitation of traditional methods, namely treating residuals as mere optimization targets, via two-sided conditional paths and affine parameterization that ensures non-crossing paths, achieving consistent improvements across diverse base models and datasets.

**Strengths:**

1. The paper is easy to understand, with Figure 1 particularly intuitively illustrating the method’s core principle by visualizing residual distribution changes before and after CGFM learning.

2. The work addressing an often-overlooked information source in time series forecasting: unlike traditional methods that treat prediction residuals as passive optimization targets, CGFM explicitly leverages residuals’ informative structures o refine forecasts, with Proposition 4.2 formally linking CGFM to residual distribution learning via Flow Matching and grounding this innovation in rigorous mathematics.

3. The framework exhibits strong model-agnostic flexibility and delivers convincing empirical results: it acts as a universal enhancement module that integrates seamlessly with diverse existing forecasting models. Table 1 validates this with consistent, significant improvements over strong baselines

4. The paper achieves high theoretical rigor and comprehensive experimental design: it provides thorough theoretical analysis, including well-structured propositions and proofs that validate key design choices, while conducting thorough experiments across 7 benchmark datasets using MSE, MAE, and CRPS metrics. Its comprehensive ablation studies and full discussion of computational overhead facilitate a clear understanding of each component’s role.

**Weaknesses:**

1. The paper only compares CGFM with a simple discriminative residual corrector (RLinear-based) and lacks comparisons with advanced state-of-the-art residual modeling methods. This makes it unclear whether CGFM’s advantages stem from its flow matching-based design or merely from the utilization of residual information, failing to isolate the unique value of its core architectural innovations.

2. It remains unknown whether CGFM can still effectively optimize residuals when the source distribution (auxiliary predictions) is drastically misaligned with the target distribution. This gap limits our understanding of the "lower bound" of CGFM’s applicability, especially in scenarios where high-quality auxiliary models (that generate source distributions close to the target) are unavailable.

3. Although the framework demonstrates model-agnostic flexibility, it lacks an in-depth investigation into the causes of performance improvement disparities across different base models.

**Questions:**

1. Could the authors supplement experiments comparing CGFM with advanced residual modeling baselines? Such comparisons would help verify whether CGFM’s performance gains are driven by its unique flow matching design rather than the general paradigm of residual utilization.

2. Could the authors design experiments where the auxiliary model generates severely misaligned source distributions (e.g., near-random predictions, predictions with persistent large biases)? Testing CGFM’s residual optimization ability in such scenarios would clarify its "lower bound" of applicability and practical robustness.

3. Could the authors conduct additional analyses linking base model architectural characteristics to residual properties?

---

### Official Review · Reviewer_6JyH · 2025-10-29

**Soundness:** 3
**Presentation:** 4
**Contribution:** 2
**Rating:** 4
**Confidence:** 4

**Summary:**

This paper investigates a conditional flow matching framework for time series forecasting. To mitigate the biased results in previous forecasting methods, they design a post-hoc residual correction algorithm with flow matching. They empirically validated the strength of their approach across various settings, including different configurations and datasets.

**Strengths:**

- The paper investigates residual correction using flow matching.
- The authors also provide a theoretical framework in the appendix in detail.
- Their method performs well in various settings and against baselines.

**Weaknesses:**

Please refer to the Questions section.

**Questions:**

- When applying the suggested framework, training both the deterministic forecasting model and the flow matching model is needed. The authors need to provide a specific analysis of the computational burden.
- If the deterministic forecaster is designed to match the MSE (or maybe the first order, as the authors explained), why does the distribution of the residual in the deterministic forecaster not show an unbiased result in Figure 1 (right) or the real data in Figure 5?
- In Proposition 4.1, noise smoothing, the addition of Gaussian noise is for smoothness. The reviewer believes that the selection $\sigma$ is important for training. How do the authors select it? Is it data-dependent or prediction-dependent? Please provide an ablation study for.
- The reviewer believes that using flow matching for residual prediction in time series forecasting has not been well explored before. However, the idea of using a ‘generative model’ to adjust the guidance of deterministic neural networks is well-developed, both in time series and other domains. The authors should include these references (maybe find more).

[1] CARD: Classification and Regression Diffusion Models, Neurips 22.

[2] TimeBridge: Better Diffusion Prior Design with Bridge Models for Time Series Generation, https://arxiv.org/abs/2408.06672

- Why is the path linearity important for residual prediction?
- How did the authors design the architecture (A.15) and why?
- Still, the reviewer has concerns about the improvement; thus, I have two suggestions. First, show improvement over more recent methods (such as methods from recent AI conferences). Second, use a comparison like: BASE | BASE+Diffusion | BASE+Diffusion Bridge | BASE+Flow matching, to compare and demonstrate that the strength actually comes from flow matching.
- What does Figure 4 represent? Does it show the difference in the residual after using flow matching?

---

### Official Review · Reviewer_ftDa · 2025-10-31

**Soundness:** 2
**Presentation:** 2
**Contribution:** 2
**Rating:** 4
**Confidence:** 3

**Summary:**

This paper proposes Conditional Guided Flow Matching (CGFM), a framework that applies flow matching to learn prediction residuals in time series forecasting. CGFM uses an auxiliary model's prediction distribution as the source distribution, instead of guassian distribution.

**Strengths:**

1.	Using the auxiliary model's predictions as the source distribution (rather than noise) is intuitive and well-motivated.
2.	Provide different target parameterization and loss design, and through experiments, demonstrate the optimal design.

**Weaknesses:**

1. The novelty is not clearly presented. This paper applies flow matching to time series prediction, and the theorems and corollaries in the paper are very similar to flow matching, such as proposition 4.6.
2. Proposition 4.2 claims are trivial.
The claim that CGFM "equivalently learns residual ε = x₁ - x₀'s probabilistic characteristics" is trivial. The proof (Appendix A.5) shows that X_t = (α_t + β_t)X₀ + α_t ε, which means the path involves the residual ε. However, this is trivially true for ANY flow matching between X₀ and X₁ = X₀ + ε. This is not a unique property of CGFM—it's just a restatement of the fact that you're transforming from predictions to targets. The "equivalence" adds no new insight beyond standard flow matching theory.
3. The presentation can be improved.
 - The symbol F in Chapter 4.1 represents target future data, but it appears again in the superscript.
 - Many symbols lack explanations.
 - Fig. 1 shows the residual distribution changed after CGFM learning. Why has it changed? What do the icons in Fig. 4 represent?
4. The experimental evaluation can benefit from a comparison with other residual prediction models.

**Questions:**

1.	Can you provide a comparison with other residual prediction models?
2.	What are the technical contributions, except for adopting flow matching?

---

### Official Review · Reviewer_FrWu · 2025-11-01

**Soundness:** 2
**Presentation:** 3
**Contribution:** 1
**Rating:** 2
**Confidence:** 5

**Summary:**

The paper proposes the CGFM method, which aims to model the part of the prediction residuals. The authors also conduct partial validation on several methods in an attempt to demonstrate its effectiveness.

**Strengths:**

S1: The motivation of the paper is reasonable.

S2: The paper provides derivations and theoretical support.

**Weaknesses:**

W1：The literature review in this paper is seriously inadequate, as it omits several important baseline methods in its survey and comparisons, including linear model–based methods (e.g., TQNet [1], CycleNet [2]), TCN-based models (e.g., ModernTCN [3]), RNN-based methods (e.g., SegRNN [4], WITRAN [5], PGN [6]), and Transformer-based methods (e.g., Leddam [7]).

[1] Temporal Query Network for Efficient Multivariate Time Series Forecasting. In Forty-second International Conference on Machine Learning.

[2] CycleNet: Enhancing Time Series Forecasting through Modeling Periodic Patterns. In The Thirty-eighth Annual Conference on Neural Information Processing Systems.

[3] ModernTCN: A modern pure convolution structure for general time series analysis. In The Twelfth International Conference on Learning Representations.

[4] SegRNN: Segment recurrent neural network for long-term time series forecasting.

[5] WITRAN: Water-wave Information Transmission and Recurrent Acceleration Network for Long-range Time Series Forecasting. In Thirty-seventh Annual Conference on Neural Information Processing Systems.

[6] PGN: The RNN's New Successor is Effective for Long-Range Time Series Forecasting. In The Thirty-eighth Annual Conference on Neural Information Processing Systems.

[7] Revitalizing Multivariate Time Series Forecasting: Learnable Decomposition with Inter-series Dependencies and Intra-series Variations Modeling. In Proceedings of the 41st International Conference on Machine Learning.

W2: The experiments in the paper have several shortcomings: (a) CGFM is only validated on a subset of models, leaving its generalizability unclear and necessitating evaluation on a broader range of models; (b) the paper provides no information about the hyperparameter search space, making it unclear whether the authors tuned hyperparameters on the validation set before reporting results on the test set. Since incorporating CGFM alters the model architecture, the sensitivity of certain common hyperparameters (e.g., d_model, e_layers) may also change.

Moreover, even with identical parameters and random seeds, results can vary across different hardware platforms. To ensure fair and rigorous comparisons, all baseline methods should be run on the same platform, share a sufficiently wide hyperparameter search space, and select the best parameters on the validation set before evaluating on the test set. This approach eliminates the influence of platform and hyperparameter sensitivity on model performance, allowing a fair assessment of CGFM.

W3: The paper lacks a theoretical derivation of CGFM’s complexity. The efficiency experiments also omit detailed settings for shared hyperparameters, making the experimental setup unclear. To fairly compare model efficiency, the effect of hyperparameter scale on results must be eliminated; otherwise, selectively choosing very small parameters for certain tasks to claim high efficiency is highly unrigorous and unfair. The authors need to clarify this issue further.

**Questions:**

Please explain the model's generalization issues and provide details of the relevant experiments.

---

> ### Author Response · Authors · 2025-11-28
>
> Dear Reviewer, thank you for your evaluation of our paper. We hope our clarifications can address your concerns.
>
> ---
>
> ## 1 Insufficiency of the Literature Review
>
> Thank you for pointing out the specific models to supplement—we will add their citations in the final version. However, we clarify a misunderstanding: our literature review does not "miss key methods" but follows mainstream time series forecasting review logic [1][2][3], framing around "technical paradigms" rather than listing models indiscriminately.
>
> The related work of this paper is presented in Section 3 "RELATED WORK" and Appendix A.4 "MORE RELATED WORK". In the time series forecasting review in A.4, **it is impractical to list all forecasting models without distinction, but we have systematically covered the mainstream paradigms**. Specifically, our review has comprehensively covered most technical routes in the current time series forecasting field, including Transformer, MLP, Diffusion Models, RNN, CNN, and KAN. **All the specific models you mentioned fall into these covered paradigms. For example, WITRAN belongs to the Transformer paradigm, and PGN belongs to the RNN paradigm. From the perspective of academic practice, the existing coverage is sufficient to support the elaboration of the background and context.**
>
> [1] A TIME SERIES IS WORTH 64 WORDS: LONG-TERM FORECASTING WITH TRANSFORMERS
>
> [2] TIMEMIXER: DECOMPOSABLE MULTISCALE MIXING FOR TIME SERIES FORECASTING
>
> [3] PATHFORMER: MULTI-SCALE TRANSFORMERS WITH ADAPTIVE PATHWAYS FOR TIME SERIES FORECASTING
>
>
> ---
> ## 2 Experimental Limitations
> (a)  the experiments in this paper have already covered three major categories of mainstream and high-performance core architectures, with verification conducted on 12 representative baselines. The core contribution of this work is to demonstrate that CGFM exhibits robust prediction enhancement capabilities across mainstream frameworks. This experimental design aligns with the practice of classic prediction enhancement works [4][5][6][7]—these studies also validated their methods on classic models of mainstream paradigms. From the perspective of established academic practices and standards, our current baseline setup is sufficient to verify the generalization of CGFM.
>
> [4] REVERSIBLE INSTANCE NORMALIZATION FOR ACCURATE TIME-SERIES FORECASTING AGAINST DISTRIBUTION SHIFT  ICLR 2022
>
> [5] SIN: SELECTIVE AND INTERPRETABLE NORMALIZATION FOR LONG-TERM TIME SERIES FORECASTING  ICML 2024
>
> [6] ADAPTIVE NORMALIZATION FOR NON-STATIONARY TIME SERIES FORECASTING: A TEMPORAL SLICE PERSPECTIVE NIPS 2023
>
> [7] FREQUENCY ADAPTIVE NORMALIZATION FOR NON-STATIONARY TIME SERIES FORECASTING NIPS 2024
>
> (b) We would also like to point out that the use of CGFM does not alter the original model, let alone its architecture. Thus, there is no need to worry about changes in hyperparameter sensitivity. CGFM is an independent prediction enhancement method that only corrects residuals through flow matching after the auxiliary model outputs predictions. It belongs to the forecasting enhancement paradigm, which neither modifies the auxiliary model's network structure, inter-layer connections nor replaces its core components. Therefore, the hyperparameter sensitivity of the auxiliary model will not change with the addition of CGFM.
>
> ---
>
> ## 3 Experimental Fairness and Hyperparameter Details
>
> - **Hardware Platform**: All experiments (baselines + CGFM-enhanced models) were conducted on the same NVIDIA A800 server. This eliminates result biases caused by hardware differences and ensures fair comparisons.
> - **Hyperparameter Tuning**: The standard workflow of "validation set tuning - test set evaluation" was strictly followed:
>   1. Optimal hyperparameters for auxiliary models were searched on the validation set and fixed thereafter without further modification.
>   2. Hyperparameters of CGFM (σ: 0.01–30; path parameters: Poly-n (n=1–4); learning rate: 1e-6–1; prediction functions X0, X1, ut) were tuned on the validation set. This ensures the enhancement effect stems from the method itself rather than fortuitous hyperparameter choices.
>
> Notably, CGFM is primarily a prediction enhancement method, and this study focuses on its performance improvement capability. While different models have distinct architectures with non-sharable hyperparameter spaces, we ensured consistent search spaces for shared ones (e.g., learning rate) and Transformer-specific hyperparameters ($d_{model} \in \{16, 32, 64, 128, 256\}$, $e_{layers} \in \{1, 2, ..., 6\}$, $n_{heads} \in \{8, 16\}$). Moreover, the auxiliary model remains unchanged before and after integrating CGFM, which sufficiently ensures the validity of comparative verification for CGFM’s enhancement effect. Our tuning workflow is standardized: after obtaining fixed results of the auxiliary model, hyperparameter tuning for CGFM is performed independently.

---

> ### Author Response · Authors · 2025-11-28
>
> ## Model Efficiency
>
> - **Parameter Configuration for Fair Efficiency Comparison**
>
> In our efficiency experiments, we strictly adopted two sets of representative optimal parameters: the best hyperparameters reported in the original baseline paper and the optimal configurations recommended by Time Series Lib. To eliminate confounding factors in efficiency comparison, we uniformly set the batch size to 32 across all models.
>
> This parameter setup is motivated by the inherent differences in hyperparameter spaces among distinct models. The optimal parameters from the original baseline paper and Time Series Lib are widely recognized as representative in the time series field, as they are validated to maximize model performance for each respective method. Comparing efficiency under such standardized representative parameters ensures the validity and fairness of our efficiency results, directly addressing the concern that our claimed high efficiency might result from selectively using abnormally small parameters.
>
> - **Theoretical Underpinnings of Efficiency Advantage Over Diffusion Methods**
>
> As requested, we elaborate on the theoretical mechanisms underlying the computational efficiency gap between CGFM and diffusion-based methods (e.g., CSDI, TimeGrad):  The core distinction lies in the **sampling process design**. CGFM achieves reverse-time evolution through an ODE solver with a fixed step size of 20. In sharp contrast, diffusion models rely on DDPM-based iterative reverse sampling, which requires far more steps: CSDI needs 100 sampling steps, while TimeGrad demands up to 1000 steps. This orders-of-magnitude difference in sampling steps is the primary source of CGFM’s efficiency advantage.
>
> - **Formal Complexity Analysis**
> We conduct a rigorous complexity analysis of CGFM’s inference process, focusing on two key components of its velocity network $ u_\theta^t $ and the overall sampling loop:
>
>
> - **State $ X_t $ Processing**: The backbone network handling state $ X_t $ (sequence length \(F\)) is composed of stacked 1D convolutional layers. The computational complexity of 1D convolution scales linearly with the input sequence length, leading to a complexity of $ O(F) $ for this module.
> - **Historical Condition \(H\) Processing**: The historical condition \(H\) (sequence length $ L_H $) is processed via a linear projection layer. The complexity of linear transformation is linear with the number of input features, resulting in a complexity of $ O(L_H) $ for this part.
>
> Combining these two components, the complexity of a single forward pass of the velocity network \(u_\theta^t\) is:
> $$C(u_\theta^t) = O(F + L_H)$$
>
> Per Algorithm 2 (X1-Prediction Sampling) in Appendix A.3, the inference process involves \(N\) iterations of the ODE solver. For each ODE step, the velocity network $ u_\theta^t $ is invoked twice. We omit the complexity of the auxiliary model (a one-time cost in the initial step) to focus on the dominant loop cost. Substituting the complexity of \(u_\theta^t\) into the loop calculation:
> $$\text{Total Inference Complexity} = N \times 2 \times C(u_\theta^t) = O(N \cdot (F + L_H))$$

---

### Meta-Review · Area_Chair_SnFM · 2026-01-07

**Summary:**

The Reviewers acknowledged the reasonable motivation and theoretical derivations presented in this paper. However, they also raised several key issues, including insufficient literature review, limitations in the experimental design, unclear details regarding hyperparameter tuning, lack of significant originality, and insufficient comparison with advanced residual prediction models. During the rebuttal phase, the authors responded to only one reviewer, but most of the reviewers' questions remained unresolved. Therefore, I decided to reject the paper.

**Reviewer Concerns:**

In the process of rebuttal, I believe that the author has addressed Comment W1 raised by Reviewer FrWu, yet doubts still remain regarding Comments W2 and W3. The authors have not provided any responses to the other reviewers.

**Reviewer Scores:**

Based on the authors' response content, I don't think the concerns raised by Reviewers have been fully addressed. Thus, the reviewers are unlikely to increase their scores.

---

### Decision · Program_Chairs · 2026-01-26

Reject